# ER O-glycosylation in synovial fibroblasts drives cartilage degradation

Le Son Tran [1,9], Joanne Chia[1,2,9], Xavier Le Guezennec [1,2], Keit Min Tham[1,2], Anh Tuan Nguyen[1,2], Virginie Sandrin[3], Way Cherng Chen[4], Tan Tong Leng[5], Sreedharan Sechachalam[6], Khai Pang Leong[7] & Frederic A. Bard [1,2,8] ✉

How arthritic synovial fibroblasts (SFs) activate cartilage ECM degradation remains unclear. GALNT enzymes initiate O-glycosylation in the Golgi; when relocated to the ER, their activity stimulates ECM degradation. Here, we show that in human rheumatoid and osteoarthritic synovial SFs, GALNTs are relocated to the ER. In an RA mouse model, GALNTs relocation occurs shortly before arthritis symptoms and abates as the animal recovers. An ER GALNTs inhibitor prevents cartilage ECM degradation in vitro and expression of this chimeric protein in SFs results in the protection of cartilage. One of the ER targets of GALNTs is the resident protein Calnexin, which is exported to the cell surface of arthritic SFs. Calnexin participates in matrix degradation by reducing ECM disulfide bonds. Anti-Calnexin antibodies block ECM degradation and protect animals from RA. In sum, ER O-glycosylation is a key switch in arthritic SFs and glycosylated surface Calnexin could be a therapeutic target.

In most tissues, fibroblasts are the key cell type producing extracellular matrices. However, fibroblasts can also degrade the matrix, allowing the turn-over of this essential component of tissues. How fibroblasts regulate these two opposite activities remains unclear.

Synovial fibroblasts (SF), also called synoviocytes are the prototypical Janus-faced cells in this regard: in healthy individuals, they contribute to the viscosity of the synovial fluid by secreting proteins such as hyaluronic acid and lubricin[1]. In arthritic diseases, they adhere and degrade the ECM of the cartilage. Understanding this change in activity during arthritis is a major focus of research in the field[2]. Most research has focused on transcriptional changes and little information is available on the regulation of protein glycosylation.

Arthritis is a group of diseases affecting diarthrodial joints[3]. Rheumatoid arthritis (RA) and Osteoarthritis (OA) are two of the most common types[4]. RA is an immune-mediated inflammatory disease, where autoreactive B cells produce autoantibodies, resulting in the formation of immune complexes in the joint[5]. This drives inflammation and the recruitment of neutrophils, macrophages and other immune cells to the area[6–9]. These immune cells secretes cytokines such as IL-1β and TNFα that activate synoviocytes. These cytokines are now therapeutically targeted, providing significant relief in many patients. OA is the most frequent and less well understood form of arthritis; it is a gradually evolving disease with a less prominent immune component[10–12]. A commonly accepted hypothesis is that mechanical damage of the cartilage leads to a low grade inflammatory condition that mediates progressive cartilage loss[11,13].

While RA and OA are different diseases, they share cellular and molecular characteristics[4,13,14]. The key pathological feature of early stage arthritis is the breakdown of cartilage extracellular matrix (ECM). This degradation is mediated in great part by the synovial membrane cells. ECM degradation eventually leads to the loss of chondrocytes, the key cells that synthesize and maintain cartilage[2,13,15].

In healthy joints, the synovial membrane surrounds and isolates the joint cavity, secreting ECM proteins in the synovial fluid. SFs are the main stromal cells of the synovial membrane, interspaced with resident macrophages[2]. During the active phases of RA, SFs alter their

[1]Institute of Molecular and Cell Biology, Singapore, Singapore. [2]Albatroz Therapeutics Pte Ltd, Singapore, Singapore. [3]Roche Pharma Research & Early Development, Innovation Center Basel, Basel, Switzerland. [4]Singapore Bioimaging Consortium, Singapore, Singapore. [5]Department of Orthopaedic Surgery, Tan Tock Seng Hospital, Singapore, Singapore. [6]Department of Hand and Reconstructive Microsurgery, Tan Tock Seng Hospital, Singapore, Singapore. [7]Department of Rheumatology, Allergy & Immunology, Tan Tock Seng Hospital, Singapore, Singapore. [8]Cancer Research Center of Marseille (CRCM), Marseille, France. [9]These authors contributed equally: Le Son Tran, Joanne Chia. ✉e-mail: frederic.bard@inserm.fr

phenotype, expressing the Fibroblast Activation Protein alpha (FAP$\alpha$) and proliferating. SF cells, as other stromal cells, express innate immune receptors. They can detect local pathogens and molecular damage, secreting cytokines that activate immune cells[16]. During inflammation, SF proliferate, forming, together with infiltrating immune cells, an enlarged synovial membrane called a pannus[17]. The pannus invades the joint cavity and degrades cartilage[13]. In particular, SF in the synovial lining layer have been shown to mediate cartilage degradation, while SF in the sub-lining tend to mediate inflammation[18]. The ECM degrading activity is due to increased production of matrix metalloproteinases (MMPs), A Disintegrin And Metalloproteinase with Thrombospondin motifs (ADAMTs) and cathepsins[19]. Arthritic SFs express both secreted (1, 3, 9) and cell surface (14, 15) MMPs[20]. MMP14 (MT1-MMP) in particular is essential for the invasive properties of SFs.

The acquisition of aberrant matrix degradation is also characteristic of SFs in OA[21]. While the OA synovial membrane typically has less immune cells than in RA, it drives cartilage degradation as in RA. What controls the switch to ECM-degradation mode of SFs is not well understood. Changes in gene expression are obviously suspected and similar transcriptional signatures have been detected in both diseases[22]. Epigenetic changes have been detected and proposed to drive the phenotype of arthritic SFs[2,23]. However, whether these alterations are sufficient to switch SFs into a degradative behavior remains unclear.

In the last few years, we found that matrix degradation by cancer cells is controlled by the upregulation of O-glycosylation of cell surface proteins. This change is induced by the intracellular relocation from the Golgi to the endoplasmic reticulum (ER) of the Polypeptide N-acetylgalactosaminyltransferases (GALNTs). These enzymes initiate the formation of O-glycans by the addition of an N-Acetylgalactosamine (GalNAc) sugar, forming the T nouvelle (Tn) glycan[24]. Tn can be detected by Tn-binding proteins such as Vicia Villosa Lectin (VVL) and Helix Pomatia Lectin (HPL)[25].

GALNTs relocation, controlled by Src and other signaling molecules, has been dubbed the GALNT Activation (GALA) pathway[25–27]. The relocation results in various substrates becoming hyper O-glycosylated, usually with clusters of glycans on neighboring amino-acids residues. This clustered glycosylation is driven by the lectin domain of GALNTs, which we found to be essential for the biological effects of GALA[25]. Based on this observation we designed a chimeric protein, composed of two GALNT2 lectin domains assembled in tandem and targeted to the ER by a KDEL sequence. This ER-2Lec construct reduces GALA induced glycosylation and inhibits biological effects induced by GALA[25].

GALA induces matrix degradation through at least two mechanisms. First, it stimulates glycosylation of MMP14, which is required for its proteolytic activity[28]. Second, GALA induces the glycosylation of the ER-resident protein Calnexin, which forms a complex with ERp57, alias PDIA3[29]. Following GALA-glycosylation, a fraction of the Cnx-PDIA3 complex is translocated to the surface of cancer cells. The complex accumulates in invadosomes and reduces disulfide bridges in the ECM[29]. This reduction is essential for the effective degradation of ECM by cancer cells[29]. Whether this glycosylation based regulation of matrix degradation is specific to cancer cells or exists also in other settings is unknown.

In this report, we show that lining SFs from arthritic patients but not from healthy individuals display marked ER relocation of GALNTs. The resulting increase of O-glycosylation, driven by re-compartmentation of the enzymes, is critical for disease progression in mice. The ER-resident protein Calnexin gets glycosylated and translocated to the cell surface where it participates in ECM degradation. We show that inhibiting Calnexin directly alleviates RA symptoms. In sum, we show that regulation of protein glycosylation by this non-transcriptional process is associated with the active phase of arthritis and could be exploited therapeutically.

## Results

### Enhanced O-glycosylation in human RA and OA synovium is driven by ER-localised GALNTs

To test whether the O-glycosylation program for ECM degradation is activated in arthritic joints, we analyzed microarrays of joint tissues by immunofluorescence using VVL and counterstained for DNA. We observed a distinct increase in Tn levels in 18/21 samples from OA patients, 2/6 samples from patients with Psoriasis arthritis (PSA) and 9/18 samples of RA patients (Fig. 1a and Supplementary Fig. 1a). We next quantified the integrated fluorescence intensity of VVL staining normalized to DNA staining. While healthy subjects samples showed little variation, enhanced Tn levels were observed in most samples of RA (average twofold increase) and OA samples (average threefold increase), with some samples displaying sevenfold increase (Fig. 1b). Analysis of patient data did not reveal a correlation between available clinical information and Tn levels. Similar results were obtained from clinical samples obtained from patients from Singapore.

To test if other glycans are affected, we stained the human tissue array with lectins Datura Stramonium Lectin, Erythrina Cristagalli Lectin (ECL) and MAL II (Maackia Amurensis Lectin). Quantification of the lectin levels in joint cores did not show significant differences between healthy individuals and patients' samples, suggesting that O-glycosylation initiation by GALNTs is specifically affected (Supplementary Fig. 1b).

Among GALNTs, GALNT2 is one of the two ubiquitously expressed enzymes[30]. We measured GALNT2 total staining intensities in the patients' joint microarrays. GALNT2 levels appeared mildly upregulated in patient samples, the increase not being statistically significant (Supplementary Fig. 1c). We investigated the intracellular localization of GALNT2, as the translocation of GALNTs to the ER has been linked to increased Tn levels[25,26,31,32]. To this end, we performed serial sectioning on a knee synovium sample from an RA patient, labeled the sections with specific markers, and conducted high-resolution confocal microscopy. Our analysis revealed that cells with high Tn levels were concentrated at the edge of the synovium, forming a layer only 2–3 cells thick (Fig. 1c). In these cells, GALNT2 colocalized with the ER marker Calreticulin, exhibiting a distinct ER localization pattern (Fig. 1c). In contrast, cells deeper in the synovium showed lower Tn levels and GALNT2 colocalized predominantly with the Golgi marker TGN46 (Fig. 1c). These findings indicate that, in arthritic synovium, cells at the periphery relocate GALNT2 from the Golgi to the ER, leading to a pronounced increase in intracellular Tn levels.

We showed previously that GALNT1 and 2 are relocalised conjointly and similarly increase Tn levels[26]. To further demonstrate that subcellular location and not expression levels of these enzymes regulate Tn levels in SFs, we compared the effect of over-expression of ER-localised versus Golgi-localised GALNT1 using available plasmids. We transformed human synovial fibroblastic SW982 cells with GFP-expressing bi-cistronic constructs with either wild-type GALNT1 (Golgi-G1) or a GALNT1 fused to an ER-retention sequence (ER-G1) as previously described (Supplementary Fig. 1d)[25,33]. All three cell lines had comparable GFP level expression (Supplementary Fig. 1e). By contrast, Tn levels measured with HPL staining showed a significant, threefold increase in ER-G1 expressing cells while GFP and Golgi-G1 cells were unaffected (Supplementary Fig. 1f)[25]. We further measured GALNT1 constructs expression levels by western blot, confirmed doxycycline induction and found comparable expression between wild type and ER-targeted GALNT1 (Supplementary Fig. 1g). The results further confirm that Tn levels are increased mostly following GALNTs subcellular relocalisation and not GALNTs expression increase. Together these results demonstrate Tn levels

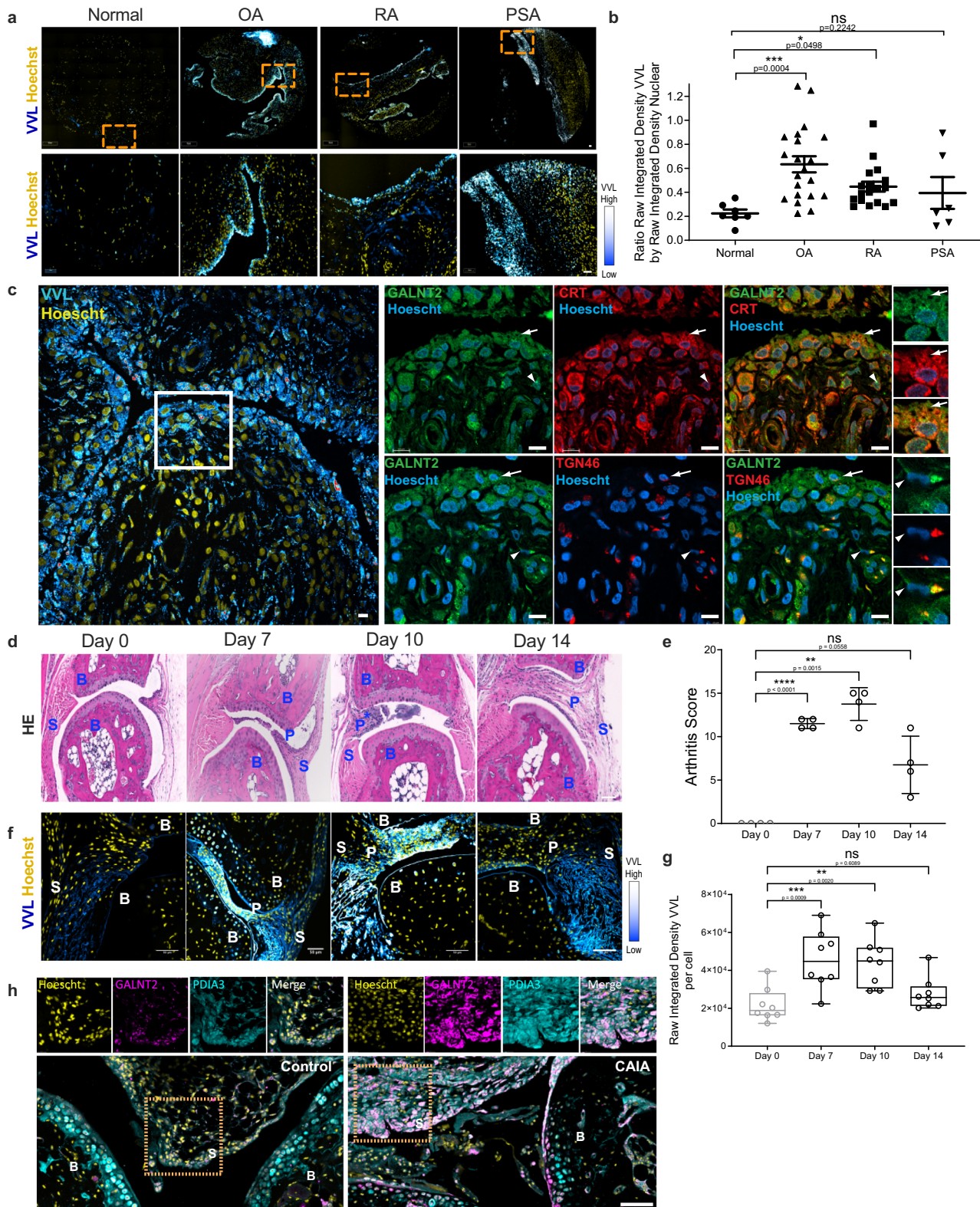

increase in arthritic joints driven by re-localisation of GALNTs to the ER.

## Arthritis symptoms correlate with Tn levels in mice

To further explore the role of O-glycosylation in arthritis, we adopted a mouse model of RA based on Collagen Antibody Induced Arthritis (CAIA)[34]. In this CAIA model, arthritis is rapidly and homogeneously induced within 1 week[34]. Briefly, animals are injected with an antibody against collagen type II, then 3 days later with lipopolysaccharide (LPS) and develop symptoms starting usually at day 5. The arthritic symptoms take about 7 days to reach a peak and last for 10 days before slowly decreasing.

Histologically, pannus invading the joint cavity was apparent on day 7 and increased in size with the influx of immune cells on day 10. By

**Fig. 1 | Enhanced O-glycosylation in arthritis synovium displays signatures of GALA activation pathway. a** Representative images of human tissue microarray (TMA) containing joint tissues from healthy subjects (Normal) and patients with osteoarthritis (OA), rheumatoid arthritis (RA) and psoriatic arthritis (PSA). (upper panel; scale bar, 50 μm), 5× magnified areas (lower panel, scale bar, 50 μm). Nuclei are stained with Hoechst in yellow and Tn glycan with Vicia Villosa lectin (VVL) with intensity color coded blue to white. **b** Quantification of Tn levels in individual cores. Data from 2 TMA slides comprising tissue sections from 21 OA, 18 RA, 6 PSA patients and 7 healthy subjects. Individual data points represent the raw integrated density of VVL staining of the core normalized to that of nuclear staining. Mean ± SEM is indicated. ∗, $p < 0.05$ ($p = 0.0498$; Normal vs RA), ∗∗∗, $p < 0.001$ ($p = 0.0004$; Normal vs OA), ns not significant ($p = 0.2242$; Normal vs PSA) (One-way ANOVA, Kruskal–Wallis test). **c** VVL staining in RA patient synovium on the left. Scale bar, 10 μm. Right panels show high magnification images (60×) in the region indicated by white box. GALNT2 was co-stained with ER marker Calreticulin (CRT) or Golgi marker TGN46 stained in red respectively. Nuclei stained with Hoescht. Arrows and arrowheads indicate cell with ER-localised GALNT2 and Golgi-localised GALNT2

respectively. Scale bar, 10 μm. **d** Hematoxylin and eosin (HE) histology images of synovial tissues sections from control (day 0) or CAIA mice at day 7, 10, and 14. S synovium, B bone, P pannus. Scale bar, 50 μm. **e** Clinical scores of four CAIA mice from day 0 to 14. Mean ± SD is indicated. ∗∗, $p < 0.01$ ($p = 0.0015$; day 10); ∗∗∗∗, $p < 0.0001$ (day 7); ns not significant ($p = 0.558$; day 14) (One-way ANOVA test). **f** Representative immunofluorescence (IF) images of synovial tissues sections stained with VVL (blue-white) and nuclei (yellow) of CAIA mice from day 0 to 10. Scale bar, 50 μm. **g** Tn levels per nuclei in the synovium of CAIA mice from day 0 to day 14. Eight joint images from four mice were quantified per time point. Whisker plots display individual values from two paws per mouse sacrificed in each time-point, boxes extend from the 25th to 75th percentiles, and error bars span max to min values. ∗∗, $p < 0.01$ (day 10, $p = 0.002$); ∗∗∗, $p < 0.001$ (day 7, $p = 0.0009$); ns not significant (day 14, $p = 0.6089$) (One-way ANOVA test). **h** Images of synovial sections from Control or CAIA mice with GALNT2 (magenta), ER protein PDIA3 (cyan) and nuclei (yellow). Scale bar, 70 μm. Zoomed images 4×; S synovial membrane, B bone.

day 14, the amount of immune cells had diminished but synovial tissue remained in the joint cavity (Fig. 1d, e).

Tn levels were evaluated by immuno-histochemistry and showed a marked increase in and around the joint in animals with marked arthritis (Supplementary Fig. 1h). We sampled joints on day 0, 7, 10, and 14 and quantified Tn levels using immuno-fluorescence. High levels of Tn were observed in the pannus invading the joint cavity at day 7 and persisted till day 10 (Fig. 1f, g). By day 14, cellular Tn levels had subsided in a large fraction of the animals, while some staining remained in fibrous material devoid of cells.

The Tn staining pattern was similar to the human samples, with a massive increase in intensity specifically in the synovium and not in other parts of the joint (Fig. 1f). GALNT2 staining was punctate and perinuclear in control joints, indicating a Golgi localisation (Fig. 1h). In arthritic joints, in the pannus specifically, GALNT2 staining was more ER-like and colocalized with the ER marker PDIA3 (Fig. 1h). GALNT2 colocalized with the ER marker PDIA3. The ER relocation of GALNTs and increased O-glycosylation in the arthritic pannus is thus conserved between species.

We next tested another model of arthritis, the collagen induced arthritis (CIA) model. DBA strain mice were induced by immunization with type II collagen injections emulsified with adjuvant[34]. These mice take about 21 days to exhibit the first signs of arthritic swelling with more severe swelling at 28–33 days. The symptoms persist up to 35 days post induction (Supplementary Fig. 1i, j). We observed pannus invasion into the joint cavity at day 28 post injection (Supplementary Fig. 1k) and reduced cartilage in the joint as revealed by Safranin O (SO) staining (Supplementary Fig. 1l). At day 35, the cartilage was almost completely excised with significant hyperplasia of cells in the joint area. High Tn levels could be seen in the cells of the invading pannus at days 28 and 35 (Supplementary Fig. 1m). These results further confirm the enhanced O-glycosylation occuring in the cells of the pannus of arthritic joints in human patients and in different animal models of arthritis.

## GALA is activated specifically in synovial fibroblasts

The synovium in RA disease is a complex tissue comprising immune cells and SFs[17]. To establish which cell type displays increased GALA, we co-stained 7d mouse CAIA joint samples with VVL, CD45 a marker of immune cells and vimentin, a marker of fibroblasts. We observed vimentin positive cells at the forefront of the invading panus (Fig. 2a). CD45 positive cells were typically clustered and located behind the invading front of the pannus. Remarkably, Tn positive cells in the pannus largely co-stained with vimentin positive region and not CD45 positive region, suggesting activation in SFs specifically (Fig. 2a). Similarly, in the CIA mice, high Tn positive cells were also vimentin positive (Supplementary Fig. 2a).

We next stained human samples of RA and OA with the Fibroblast Activated Protein alpha (FAPα), a marker of synovial lining fibroblasts[20]. The FAPα positive cells formed a layer located at the edge of the synovium, with a larger layer of CD45 positive cells behind in the RA sample (Fig. 2b, c). In OA samples, the number of CD45 cells was reduced but FAPα cells similarly displayed high VVL staining (Supplementary Fig. 2b). Strikingly, VVL staining co-localised exclusively with FAPα in both RA and OA conditions.

In sum, SFs lining the edge of the synovium are the major cells displaying high Tn and GALNTs ER relocation in OA and RA samples.

## Cytokines and ECM stimulates GALA in SF

To explore how GALNTs relocation is activated in arthritic conditions, we used primary human SFs derived from patients. FACS analysis using CD90 and CD45 as markers of SFs and immune cells respectively established the >90% purity of the cell preparations we used (Supplementary Fig. 2c). We also performed high content imaging of these SF patient derived cells for Tn levels. Seeding cells on plastic wells showed increased levels of Tn in OA and even more in RA cells as compared to healthy control SF (HCSF) (Fig. 2d, e).

Next, we stimulated SFs with TNFα and IL1β cytokines, which drive disease progression in RA[35]. Individual cytokines had a relatively limited effect on GALA activation, however the combination of both cytokines (labelled CYTO) induced a twofold increase in Tn levels. Interestingly, the effect was marked in RASF, more limited in OASF and almost nonexistent in HCSFs, suggesting arthritic SF are primed to activate GALA in response to these cytokines.

Next, we tested whether ECM proteins might activate SF. Exposing SF to cartilage or rat tail ECM activated GALA by a striking threefold in both OASF and RASF (Fig. 2d, e). By contrast, HCSF cells were almost non-responsive. The combination of CYTO and ECM had an additive effect on Tn levels (Fig. 2d, e). Tn staining was exclusively detected in the Golgi of unstimulated cells, but displayed marked ER-like pattern in stimulated OASF and RASF (Fig. 2e). The striking difference with healthy control cells (HCSF) suggest again that RASF and OASF are primed to activate GALA with some ECM proteins.

Similar to the primary SFs, stimulation with CYTO and cartilage ECM in human synovial cell line SW982 resulted in increases in Tn levels (Supplementary Fig. 2d, e). GALNT2 was co-stained with ER (PDIA4) and Golgi (TGN46) markers. While GALNT2 signal is strictly at the Golgi in control cells, a fraction colocalises with ER marker PDIA4 in stimulated cells (Fig. 2f). We quantified the level of colocalisation between ER marker PDIA4 and GALNT2 using previously published analysis method on ImageJ[32]. Manders coefficient M2, the fraction of PDIA4 overlapping with GALNT2, was increased by about 3.7 fold in stimulated cells compared to untreated control (Fig. 2g).

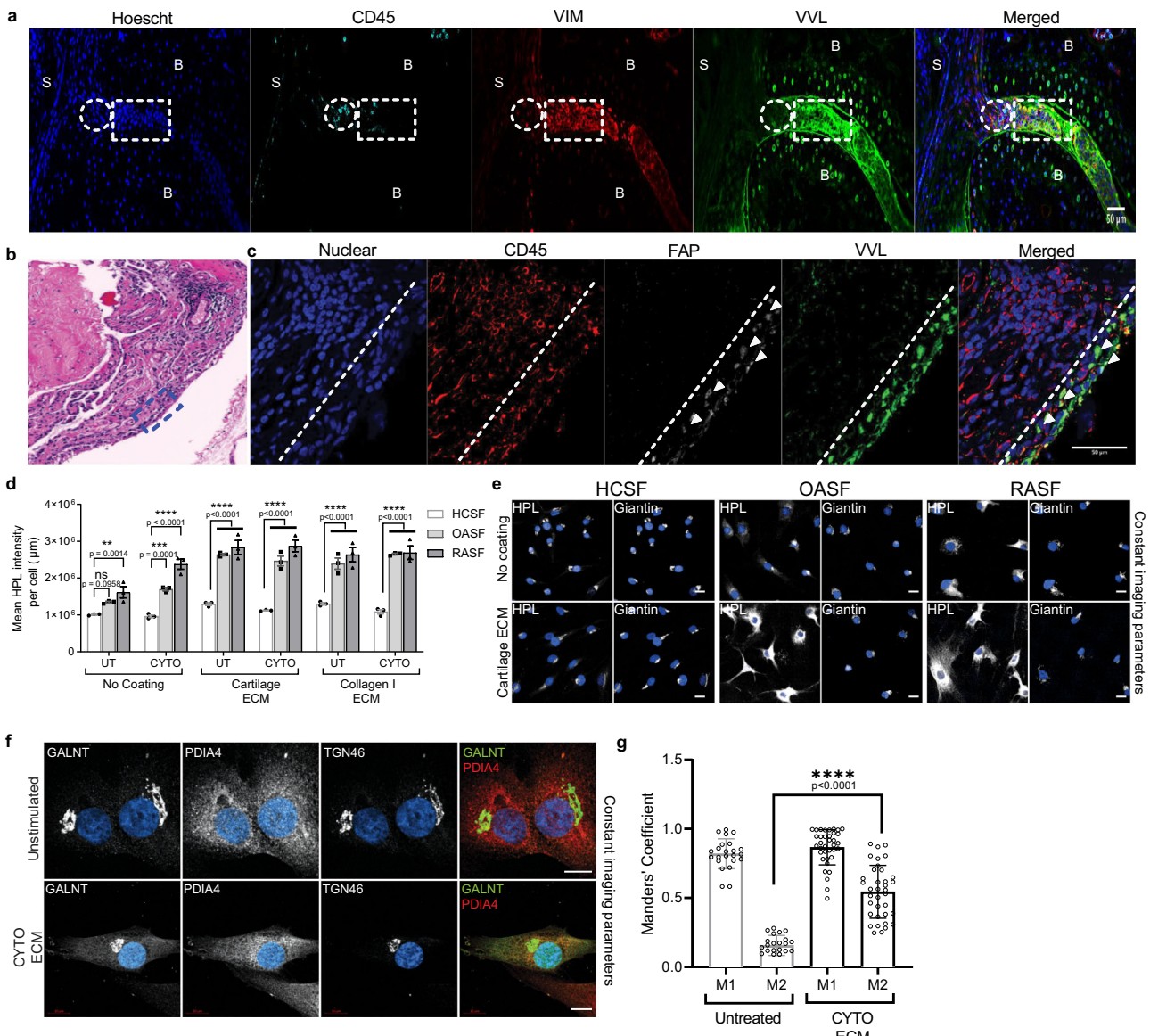

**Fig. 2 | Synovial fibroblasts (SF) are the major cell type displaying GALA in arthritic synovium. a** Co-staining of VVL, mouse fibroblast marker Vimentin (VIM), immune cell marker CD45 and nuclei in the synovium from 7-day CAIA mice joint section, relative enrichment areas for immune cells (circle) and fibroblasts (rectangle) in the pannus tissues is indicated. Scale bar, 50 μm. S synovium, B bone. **b** Representative HE image of synovial tissues obtained from RA patients. *: Infiltration of immune cells in the sub-lining of RA synovium; SL synovial lining. Scale bar, 100 μm. **c** Representative IF images of human RA synovium from (**b**); lining synovial fibroblasts (SF) identified by FAPα (arrowheads) and immune cells identified by CD45 in the sub-lining (limit of lining indicated by the dashed line). Scale bar, 50 μm. **d** Quantification of *helix pomatia* lectin (HPL) levels per cell in patient purified HCSF, OASF and RASF under basal conditions and upon presence of cytokines IL1β and TNFα (CYTO) plated on plastic (No coating) or plated on cartilage ECM or collagen I enriched ECM. Data are the mean ± SEM of three

independent experiments. **, $p < 0.01$ ($p = 0.0014$; No coating, UT HCSF vs RASF) ***, $p < 0.001$ ($p = 0.0001$; No coating, CYTO, HCSF. vs OASF), ****, $p < 0.0001$, NS not significant ($p = 0.0958$; No coating, CYTO, HCSF vs OASF) (Two-way ANOVA test). **e** Representative immunofluorescence images of HCSF, OASF, and RASF stained with HPL, Golgi marker Giantin and Hoechst on plastic wells or cartilage ECM treated wells. Scale bar, 20 μm. **f** Representative images of wildtype SW982 cells stimulated with or without CYTO and cartilage ECM stained with GALNT2, ER marker PDIA4 and Golgi marker TGN46. Images were acquired at 100× magnification. Scale bar, 10 μm. **g** Quantification of Mander's coefficient of GALNT1 and ER marker PDIA4 in (**f**). M1 represents the fraction of GALNT1 staining coincident with the ER and M2 represents the fraction of the ER marker coincident with GALNT1 staining. 23 and 36 cells from unstimulated and CYTO-cartilage ECM stimulated respectively. Data present are the mean ± SD of cells quantified from two experiments. ****, $p < 0.0001$ (Two sided Student's *t*-test, two-tailed *p*-value).

To summarize, our analyses revealed that OA and RA patients' SFs have elevated GALA levels compared to healthy SFs in cell culture. RASF activate GALA in response to cytokines, while both RA and OASF activate it upon exposure to ECM proteins. By contrast, healthy SFs had very limited responses, suggesting the glycosylation pathway is primed for activation in patients' cells.

## GALNT inhibitor prevents arthritis symptoms in vivo

Current small molecule inhibitors of GALNTs do not allow for ER-specific inhibition of GALNTs. To achieve this goal, we turned to the ER-specific GALNTs inhibitor ER-2Lec, which we described previously[25]. Briefly, this inhibitor is built from the lectin domain of GALNT2, which binds Tn. The GALNTs lectin domain mediates

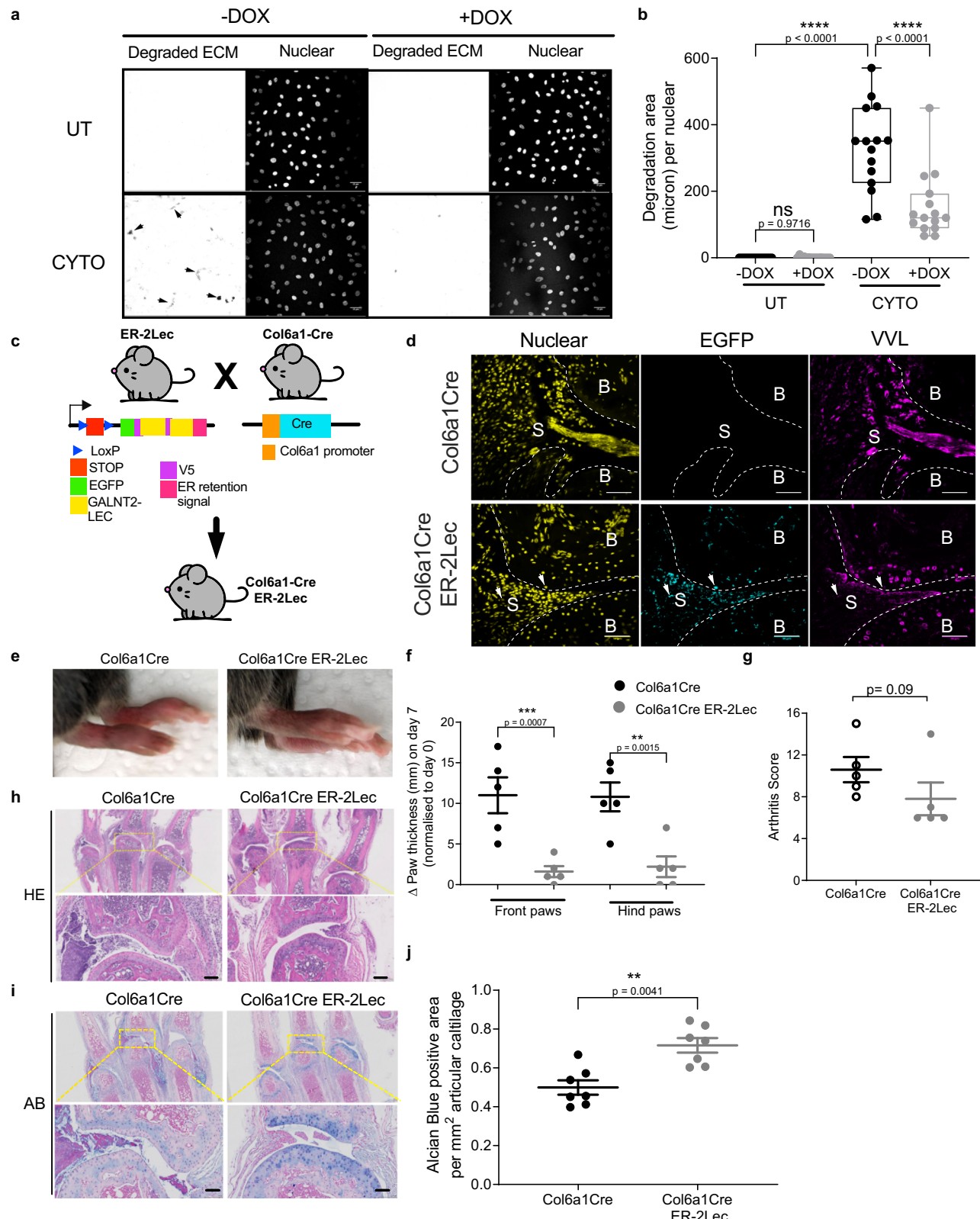

**a**

**b**

**c**

**d**

**e**

**f**

**g**

**h**

**i**

**j**

clustered glycosylation (multiple adjacent Thr or Ser residues modified); ER-2Lec is thought to compete with this process, it reduces surface Tn levels in GALA positive cells (Gill et al.[25]).

To test the effect of ER-2Lec on cartilage degradation, we sought to establish a stable, inducible cell line. The human synovial sarcoma SW982 cells are a good model for arthritic SFs as they degrade collagen efficiently. We generated a stable SW982 cell line with ER-2Lec under a

doxycycline inducible promoter system. ER-2Lec expressing cells stimulated with CYTO and cartilage ECM displayed virtually no increase in Tn contrary to SW982 GFP control cells (Supplementary Fig. 3a, b).

To measure cartilage ECM degradatory properties of these cells, we used a sandwich assay with fluorescent gelatin previously described[29]. Similar to SFs in RA disease, SW982 cells stimulated with CYTO were more active at degrading the ECM (Fig. 3a, b). However,

**Fig. 3 | GALA activation in SF drives cartilage damage in arthritis mice.**
**a** Representative images of cartilage matrix degradation activity of SW982 SF cells expressing doxycycline (DOX) inducible 2 Lectin domains of GALNT2 in the ER (ER-2Lec). ER-2Lec SW982 cells were grown on untreated (UT) coverslips or with cartilage ECM and were stimulated with RA associated cytokines (CYTO). Arrows show degraded fluorescent regions. **b** Quantification graph of matrix degradation activity presented in (**a**). Data correspond to the mean ± SEM for three independent experiments. Each data point represents the total degraded area (µm) per nuclei per well. ****, $p < 0.0001$ (UT vs CYTO in −Dox condition and −Dox vs +Dox in CYTO treatment) (One-way ANOVA test). **c** Schematic diagram of transgenic strategy: ER-2Lec transgenic mice expressing 2 Lectin domains of GALNT2 in the ER under the control of 2 LoxP stop sites (ER-2Lec) are crossed with another transgenic line expressing Collagen type VI promoter Cre (Col6a1Cre) to obtain the offspring mice (Col6a1Cre ER-2Lec). **d** Representative synovium immunofluorescence images of Col6a1Cre ER-2Lec or control Col6a1Cre mice stained with DAPI, GFP, and VVL. Positive GFP-ER-2Lec expression is indicated with arrowheads. B bone, S synovium. Scale bar, 50 µm. **e**–**g** Arthritis symptoms comparison; (**e**) representative photographs, note the difference in redness, (**f**) change in paw thickness measurements and (**g**) clinical scores in Col6a1Cre control and Col6a1Cre ER-2Lec mice at day 7 post arthritis. Data are the mean ± SEM of two 2 independent experiments, $n = 5$ mice per group. In (**f**, **g**), **, $p < 0.01$ ($p = 0.0015$; Col6a1Cre vs Col6a1Cre ER-2Lec, Hind paws); ***, $p < 0.001$ ($p = 0.0007$; Col6a1Cre vs Col6a1Cre ER-2Lec, Front paws) (Two sided Student's $t$-test). **h**, **i** Histological analysis of synovium at day 7. **h** Representative histology images of HE and (**i**) Alcian blue (AB) stainings. Lower panels, 5× magnified areas. Scale bar, 100 µm. **j** Quantification of AB positive staining areas of synovium. Each data point represents an average of positive staining area per mm² articular cartilage from different metacarpophalangeal joints of five animals. Data are shown as mean ± SEM. **, $p < 0.01$ ($p = 0.0041$; Col6a1Cre vs Col6a1Cre ER-2Lec) (Mann–Whitney test, two-tailed $p$-value).

when ER-2Lec expression was induced, the degradation was reduced by more than twofold (Fig. 3a, b).

Encouraged by these results, we generated a transgenic mouse line with a ER-2Lec ORF in a Lox cassette. We crossed them with mice expressing Cre under the collagen VI alpha1 (Col6a1) promoter (Fig. 3c). Collagen VI is expressed by joint mesenchymal cells, and in particular SFs[36,37]. This genetic cross is expected to result in ER-2Lec being expressed mostly in SFs in the joints. We induced CAIA in these mice and monitored ER-2Lec-GFP expression and Tn levels in the pannus after 7 days. Tn was markedly reduced, consistent with GALA inhibition (Fig. 3d).

We monitored RA symptoms in Col6a1Cre ER-2Lec animals and in Cre expressing controls. The ER-2Lec expressing animals displayed a marked reduction of swelling in the paws (Fig. 3e and g). We also used the internationally defined arthritis score in a blinded evaluation and observed a consistent reduction of symptoms in Col6a1Cre ER-2Lec animals (Fig. 3f).

We also performed histological analysis on day 7 using H&E and the alcian blue (AB) and safranin-O (SO) stains, which are commonly used to reveal the cartilage fraction of joints. H&E staining of Col6a1Cre ER-2Lec joints showed a reduction of pannus size, consistent with the reduction of swelling (Fig. 3h). Interestingly, the infiltration of immune cells seemed much reduced in the ER-2Lec expressing animals (Fig. 3h, i and Supplementary Fig. 3c). The AB positive region was also significantly preserved in CAIA Col6a1Cre ER-2Lec animals (Fig. 3i, j). Significant change was also obtained after quantification of SO stains (Supplementary Fig. 3c–e). Taken together, our results demonstrate that inhibiting ER-localised GALNTs with the ER-2Lec protein in SF can limit arthritis disease progression.

## CNX is glycosylated and cell surface exposed in arthritic SF

We recently described Calnexin (CNX) as a glycosylation target and effector of ER-localised GALNTs[29]. Upon glycosylation, CNX is translocated at the cell surface and, in conjunction with PDIA3, mediates the cleavage of disulfide bonds in ECM proteins. This reductive activity is essential for matrix degradation by cancer cells[29]. In SW982 cells, we found that CNX is hyper-glycosylated by ~sixfold after stimulation with cytokines and ECM (Fig. 4a, b). This glycosylation was dependent on GALNTs relocation as expression of ER-2Lec was able to significantly reduce CNX glycosylation (Fig. 4a, b). In addition, we found, using FACS, that CNX surface expression increased significantly by about 10% after stimulation of SW982 SF cells with CYTO and ECM (Fig. 4c, d). Strikingly, the increase in cell surface CNX signal was fully repressed by ER-2Lec expression (Fig. 4c, d).

We sought to confirm this result in primary cells obtained from patients. In SFs from healthy controls (HCSF), the proportion of cell surface CNX positive cells was only ~7% and stimulation with cytokines and ECM increased it slightly (Fig. 4e, f). By contrast, SF cells from the patient suffering from RA (RASF) or OA (OASF) displayed significantly

increased levels and were more sensitive to stimulation, with a three-fold increase in the percentage of cells with cell surface CNX (Fig. 4e, f). Overall, these results indicate that CNX glycosylation and its cell surface exposure is enhanced in arthritic SFs and is dependent on GALNTs relocation.

## Anti-CNX antibodies prevent arthritic symptoms in CAIA mice

We have previously shown that antibodies against CNX can block ECM degradation by preventing the reduction of disulfide bonds[29]. We hypothesized that blocking Calnexin would similarly prevent cartilage ECM degradation. We tested first for the presence of disulfide bonds in cartilage ECM using a previously described method[29]. Briefly, cartilage ECM was either untreated or reduced with TCEP, then exposed to N-ethylmaleimide (NEM) and the NEM modified cysteines were subsequently detected by the anti-OX133 antibody. We observed an abundant OX133 signal colocalizing with collagen III (COL III)/COL I and fibronectin (FN)/COL I fibers, suggesting cartilage ECM is heavily cross-linked with disulfide bonds (Fig. 5a).

We tested the effect of anti-CNX antibodies on ECM degradation by seeding overnight OASF cells on cartilage ECM covered with fluorescent gelatin. A polyclonal anti-CNX antibody completely blocked degradative activity (Fig. 5b, c).

To examine the antibody's ability to block disulfide bond reduction, SW982 ER-G1 cells were sparsely seeded on cartilage ECM and treated with either isotype or anti-CNX antibody for 2 days. The preparation was then treated with NEM to detect reduced cysteine residues using the OX133 antibody[29]. Control cells showed strong OX133 signal around collagen I fibers (Supplementary Fig. 4a). In contrast, the presence of anti-CNX antibody significantly reduced OX133 labeling (Supplementary Fig. 4a). Quantification revealed approximately a 60% reduction in the intensity of OX133 signal in collagen I fibers surrounding the cells with anti-CNX antibody Supplementary Fig. 4b, c).

Building on these promising findings, we proceeded to administer the anti-CNX antibody to animals. In a weight monitoring study involving three antibody injections over a 10-day period, no weight loss was observed (Supplementary Fig. 4d).

Subsequently, we treated CAIA animals with a polyclonal rabbit anti-CNX antibody (ab22595) by administering 25 micrograms every two days from day 3 to day 7 after CAIA initiation (Fig. 5d). Paw thickness was regularly monitored, and arthritic scores were measured at day 10. Remarkably, the paws of anti-CNX treated animals displayed reduced swelling compared to control animals treated with an isotype antibody (Fig. 5e, f). Although some redness and swelling in the fingers persisted, contributing to an elevated arthritic score, the average score of treated animals was half that of the CAIA control animals (Fig. 5g). These results are reminiscent of the outcomes observed with ER-2Lec expression.

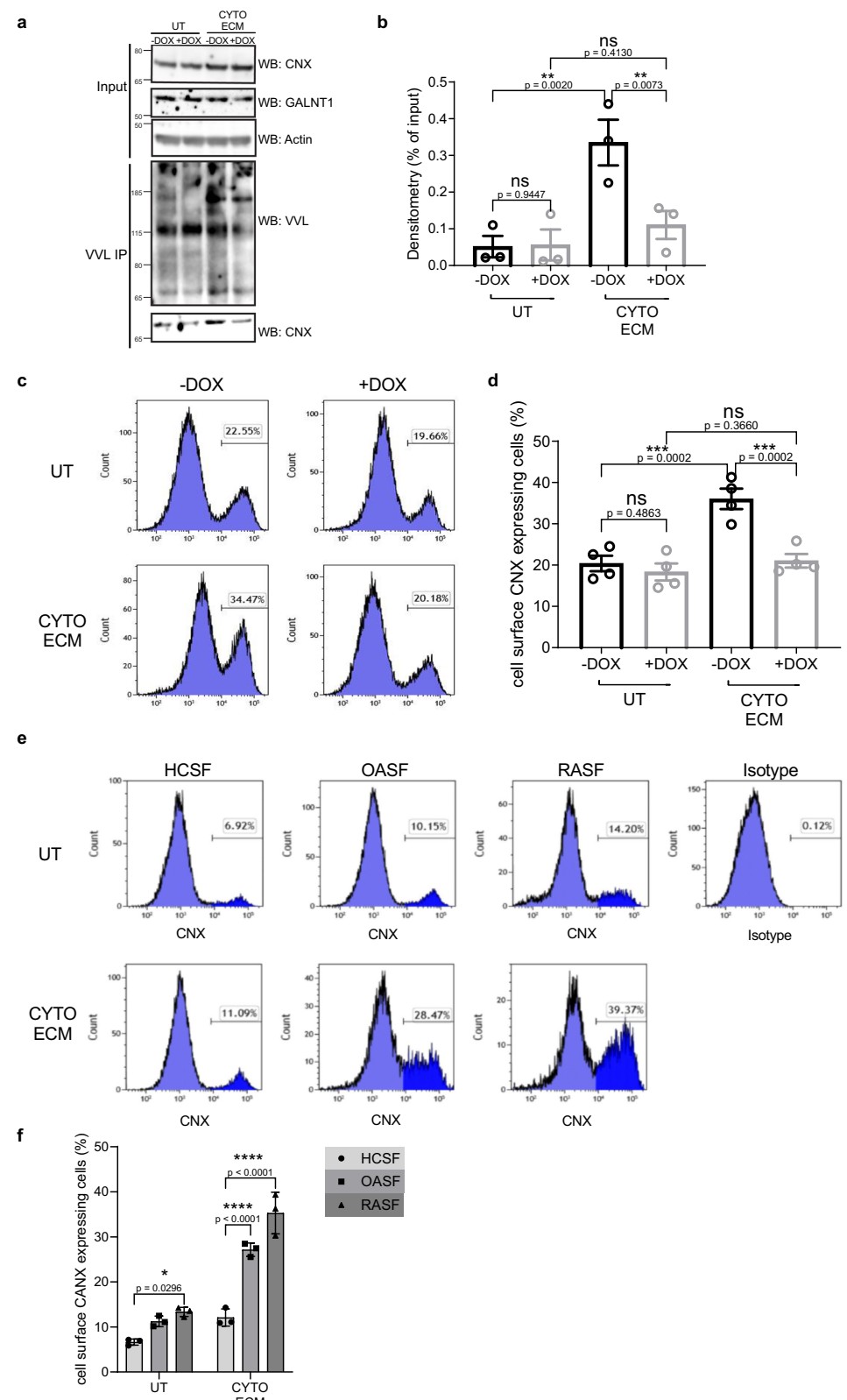

Control animals at day 10 exhibited a significant reduction in AB and SO positive cartilage at the histological level, along with synovium adhesion to the underlying bone, indicating potential bone remodeling (Fig. 5h, i and Supplementary Fig. 4e, f). In contrast, anti-CNX treated animals displayed a well-preserved joint with abundant remaining cartilage (Fig. 5h, i and Supplementary Fig. 4e, f).

Our working hypothesis suggests that the anti-CNX antibody directly inhibits lining SFs. To verify whether the antibody indeed interacted with these cells, we stained the joints of treated animals using an anti-rabbit IgG. Clear signal detection was observed in the synovial cells of animals treated with anti-CNX antibody, while no signal appeared in animals treated with a control rabbit IgG (Fig. 5j).

**Fig. 4 | GALA activates O-glycosylation and cell surface exposure of CNX in arthritis primed SF. a** VVL lectin pulldown and CNX co-precipitation with extract from SW982 ER-2Lec inducible cells, untreated (UT), or subjected to stimulation with arthritis associated cytokines (CYTO) and cartilage ECM. Actin was used as a loading control. **b** Quantification graph of western blot. Data are shown as mean ± SEM and representative of three independent experiments. **, $p < 0.01$ ($p = 0.002$; UT vs CYTO ECM, −dox condition and $p = 0.0073$; −dox vs +dox, CYTO ECM treatment); ns not significant ($p = 0.413$; UT vs CYTO ECM, +dox condition and $p = 0.9447$; −dox vs +dox, UT) (One-way ANOVA test). **c** Representative flow cytometry histogram plots with SW982 ER-2Lec inducible cells, $x$-axis depicts CNX cell surface positive signal and $y$-axis marks cell counts. **d** Quantification graph of percentage of cells expressing cell surface CNX. Data are shown as mean ± SEM

from four replicate wells from two independent experiments. ***, $p < 0.001$ ($p = 0.0002$; UT vs CYTO ECM, −dox condition and $p = 0.0002$; −dox vs +dox, CYTO ECM treatment); ns, not significant ($p = 0.366$; UT vs CYTO ECM, +dox condition and $p = 0.4863$; −dox vs +dox, UT) (One-way ANOVA test). **e** Flow cytometry histogram plots with patient derived SF untreated (UT), or subjected to stimulation with arthritis associated cytokines (CYTO) and cartilage ECM, $x$-axis depicts CNX cell surface positive signal and $y$-axis marks cell counts. A control surface stain with isotype antibody is presented. **f** Quantification graph of percentage of cells expressing cell surface CNX. Data are shown as mean ± SEM of three replicate wells, representative of two independent experiments. *, $p < 0.05$ ($p = 0.0296$; UT HSCF vs RASF); ****, $p < 0.0001$ (CYTO ECM HSCF vs OASF or RASF) (One-way ANOVA test).

To validate our findings, we employed a different antibody format, a human derived single chain Fv (scFv), which specifically targeted the lumenal domain of CNX and was developed in our lab. The specificity of the anti-CNX scFv was confirmed through western blotting (Supplementary Fig. 5a) and immunofluorescence using CNX-mCherry expressing cells Supplementary Fig. 5b). Subsequently, CAIA mice were treated with the scFv by injecting 100 micrograms every two days from day 3 to day 9 after CAIA initiation (Supplementary Fig. 5c). Similar to the results observed with anti-CNX IgG1 (Fig. 5f), significant relief of arthritis symptoms was observed in animals treated with anti-CNX scFv compared to the control group (Supplementary Fig. 5d, e). Histological stainings further revealed detectable scFv signal in the synovial lining of the joints, co-localizing with the fibroblast marker vimentin (Supplementary Fig. 5f).

Overall, these results indicate that inhibiting CNX results in a potent inhibition of cartilage ECM degradation and could form the basis of arthritis therapeutics.

## Discussion

In this study, we show that arthritis joints display an increase of the Tn O-glycan levels. The phenomenon is observed in several types of arthritis, including RA and OA and is induced in mouse models of RA.

These high Tn levels are due to the GALA pathway, the induced intracellular relocation of GALNTs from Golgi to ER. This relocation is observed directly in patient and mouse tissue samples. Indeed we find that only relocation and not overexpression of GALNTs significantly affect Tn levels in SFs. A likely explanation is that relocated GALNTs modify new substrates such as ER-resident proteins and that these Tn glycans are not capped by enzymes such as C1GALT[27].

The relocation occurs in cells that have markers of activated SFs, located at the edge of the synovial membrane, in close proximity with the cartilage. This suggests that specific signals stimulate the relocation of GALNTs in synovial fibroblats. Previously, relocation has been found to be stimulated by Src or EGFR activation[26,32]. Other signaling molecules, such as the ERK8 kinase in HeLa cells, inhibit the relocation[31]. Here, we found that SFs activate GALNTs relocation in response to an IL-1β and TNF-α cytokine mixture. These cytokines drive RA symptoms. GALA activation is more intense in RA patient derived fibroblasts than in healthy human SFs, suggesting RA SFs have been primed to respond to these cytokines. How these cytokines induce GALNTs relocation is unclear, however IL-1β has been reported to activate the tyrosine kinase Src[38].

Exposure to ECM strongly activates GALA in both OA and RA SF, but not in cells from healthy joints. It has been proposed previously that elements of cartilage ECM activate SF and are involved in arthritis development[39]. Injection of fibronectin in joints leads to the degradation of cartilage proteoglycans[40]. Integrins, the fibronectin receptors, are also activators of the Src kinase[41,42]. Thus, a signaling cascade might link external ECM fragments to integrins, Src and then GALNTs relocation, activating ECM degradation[26]. This hypothetical cascade would feed a pathological positive feedback loop.

On the other hand, in the mouse model of CAIA, there is a reduction in cells with high Tn at day 14. GALA is a reversible event, for instance GALNTs flow back to the Golgi after Src stimulation is withdrawn[31]. This suggests that the pathway could be activated transiently in physiological conditions, whereas activation is sustained in pathological conditions due to continuous signalling. SFs from arthritic joints have been proposed to be epigenetically primed for degradation[15]. Indeed, OASF and RASF display comparable global methylation profiles, which are distinct from SF from healthy subjects, with differences in genes involved in PDGF and EGF signaling, regulators of GALNTs relocation[23,32]. Epigenetic priming could induce the higher propensity of arthritic SFs to activate GALNTs relocation.

In patient samples, Tn is detectable in all samples of OA, RA, and PSA, but the levels are variable. This could have various causes, including individual clinical differences. An interesting hypothesis is that GALA might only be fully activated during the active ECM degradative phase of the disease, a phase that could correspond to flares in patients. By contrast, in phases of remission, there is probably less ECM degradation and correspondingly lower glycosylation levels. Overall, it will be important to expand the range of patient samples being analyzed and test whether some clinical parameters such as rate of disease progression might correlate with Tn levels.

GALA glycosylation is probably synergistic with other regulatory mechanisms. For instance, we found some increased levels of GALNT2 in the synovial tissues of arthritic mice, consistent with gene expression data reported in previous studies[43,44]. The substrates of GALA might also be regulated by other mechanisms.

Among these substrates is MMP14, a cell surface protease that degrades collagen fibers and activates other MMPs[28,45]. MMP14 is one of the MMPs involved in arthritis and activates MMP-2 and 13[46]. MMP14 O-glycosylation is essential for full collagenase activity and occurs in a low complexity region of the protein in the form of a cluster: six or more amino-acids are modified with GalNAc or more complex O-glycans[28].

Clustered glycosylation is a frequent feature of GalNAc glycosylation, exemplified in mucin proteins. The ER-2Lec chimeric protein inhibits this clustered glycosylation[25]. As in cancer cells, ER-2Lec reduced the levels of Tn signal in SFs. It is likely to inhibit at least partially MMP14 and CNX glycosylation and thus inhibit ECM degradation by SFs in vitro and in vivo. As GALNTs act on thousands of proteins and preliminary, unpublished data suggest that their relocation affects many proteins, additional glycoproteins may be involved in the pathological activity of SFs and affected by ER-2Lec[47]. Expression of ER-2Lec in collagen type VI expressing cells results in protection from the loss of cartilage in CAIA-treated mice. In the joint, ER-2Lec expression was mostly restricted to SFs, with no expression detected in immune cells.

CNX also displays a clustered glycosylation pattern, located in the N-terminal region[29]. As shown in a previous study, CNX participates in the reduction of disulfide bonds in liver ECM proteins[29]. Cartilage ECM contains an abundance of disulfide bonds. This disulfide crosslinking, like other cross-linking bonds, likely prevent the action of proteases[48].

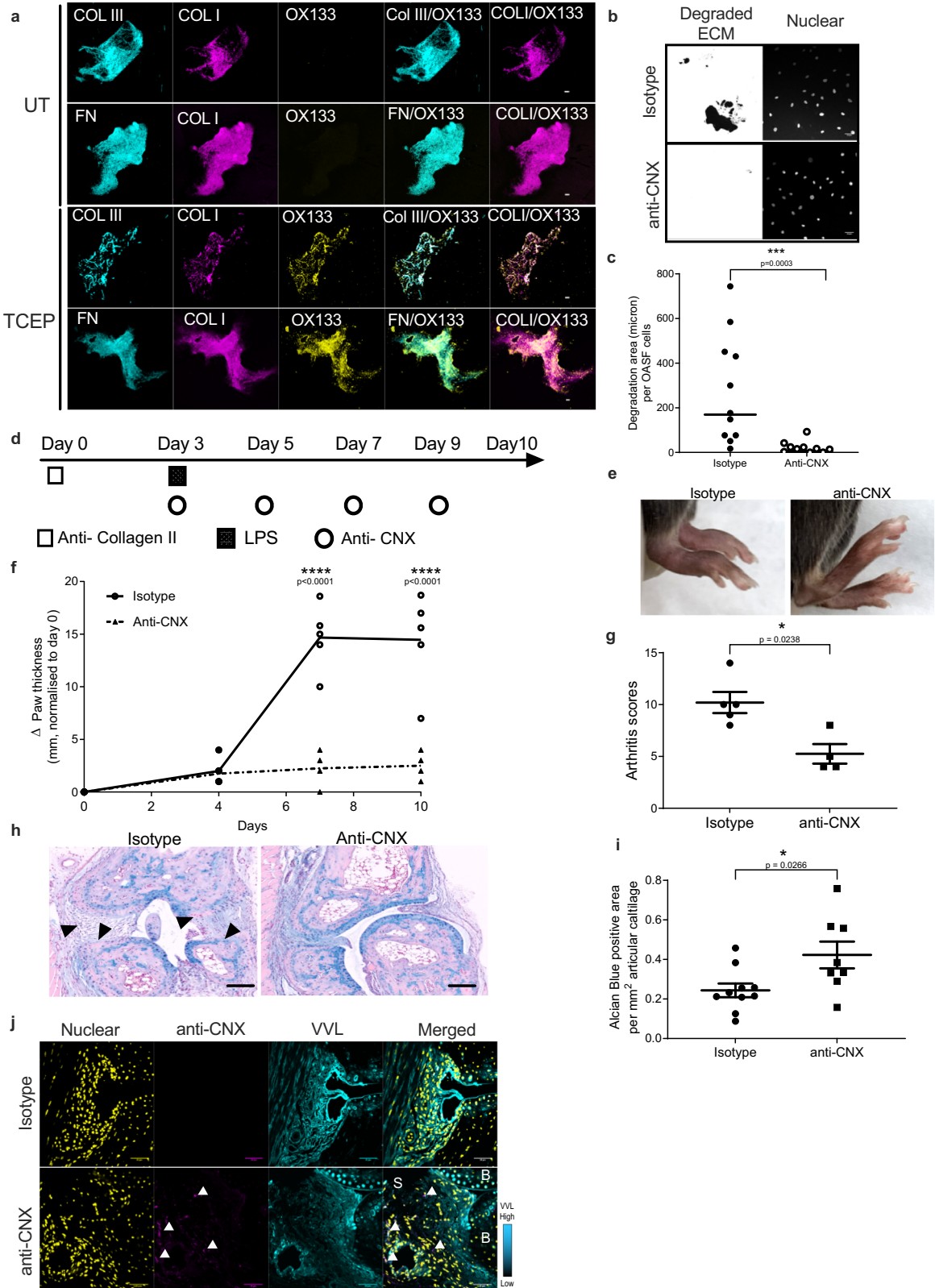

Anti-CNX antibodies block matrix degradation by SFs in vitro. In vivo, the antibodies provide a significant protection of cartilage ECM and arthritic symptoms in CAIA animals.

This potential arthritic treatment is obviously still at very early stages. A polyclonal rabbit antibody and a fragment scFv antibody were used and the animals were treated at early time points in the development of the symptoms. It remains to be establish what effect a monoclonal antibody would have on a full blown arthritic crisis.

Other strategies to inhibit synoviocytes have been developed, such as targeting the adhesion molecule Cadherin 11[49,50]. More recently, targeting the tyrosine phosphatase PTPRS at the cell surface of SFs has also been shown to protect the cartilage in RA mice[51]. In

**Fig. 5 | Blockage of CNX protects CAIA mice from cartilage degradation and development of severe arthritis. a** Representative IF staining images show abundant presence of collagen fibres containing collagen type III (Col III), collagen type I (Col I) and fibronectin in cartilage ECM. Collagen disulfide bonds are chemically reduced using TCEP which can be detected by staining with OX133 antibodies or left untreated (UT). Scale bar, 5 µm. **b** Representative images and (**c**) quantification graph show decreased matrix degradation activity in primary fibroblasts isolated from OA patients after 48 h incubation with 5ug anti-CNX antibodies (ab22595, abcam) as compared to those treated with isotype control antibodies (ab37415, abcam). Data correspond to the mean ± SEM and representative data for three independent experiments. Each data point represents the total degraded area (µm) per nuclear per well. ***, $p < 0.001$ ($p = 0.0003$) (Mann–Whitney test, two-tailed $p$-value). **d** Antibody treatment scheme. Mice were intraperitoneally injected with 2 mg anti-collagen II antibodies (Chondrex Inc.) on day 0, followed by 50 µg lipopolysaccharide (LPS) injection on day 3. Antibody treatment was subsequently administered at 25 µg of antibody per mice on the same day and the treatment was repeated every two days. A total of four antibody treatments was given for each treated mouse. **e** Representative photographs of CAIA mice treated with anti-CNX antibodies or isotype antibodies at day 10. **f** Paw thickness measurement in CAIA animals after injecting with anti-CNX antibodies. Data represent mean ± SEM, $n = 5$ mice injected with isotype antibody and $n = 4$ mice injected with anti-CNX antibody. ****, $p < 0.0001$ (day 7 and day 10) (Two-way ANOVA test). **g** Clinical scores of CAIA mice at day 10 following treatment with isotype control antibodies ($n = 5$) or anti-CNX ($n = 4$) antibodies. Data are mean ± SEM. *, $p < 0.05$ ($p = 0.0238$) (Mann–Whitney test, two-tailed $p$-value). **h, i** Representative histological images (**h**) and quantification (**i**) of Alcian blue (AB) staining (arrow bars) in CAIA mice treated with isotype control ($n = 5$) or anti-CNX antibodies ($n = 4$). Individual data points represent average of positive staining area per mm$^2$ articular cartilage from two individual metacarpophalangeal joints of each animal. Data are shown as mean ± SEM. *, $p < 0.05$ ($p = 0.0266$) (Mann–Whitney test, two-tailed $p$-value). **j** Representative IF images show co-staining of anti-CNX antibodies (arrow heads) and VVL in CAIA mice injected with anti-CNX antibodies ($n = 4$) or isotype control antibodies ($n = 5$) at day 10. B bone, S synovium. Scale bar, 50 µm.

addition, the targeting of MMPs has been researched for several decades and specific MMP inhibitors such as Trocade have demonstrated protective effects against RA and OA in animal models[52,53]. Poor tolerability of these compounds led to clinical trial failures[54]. To date, while some progress has been made, inhibiting MMP therapeutically remains relatively challenging[55].

In conclusion, our data shows that the GALA pathway, GALNTs activation by relocation to the ER, is a critical switch for cartilage degradation in arthritic SFs. These findings open perspectives for biomarker discovery to recognise arthritis active stages. The targeting of surface CNX by antibodies in these SFs represent a potential new therapeutic approach against this family of diseases.

## Methods

### Patient samples, cell lines and mouse strains
Patient samples: Human tissue microarray (Catalog number 401 3201) of synovial tissues was obtained from provitro AG (Berlin, Germany). Patients' information were provided in the datasheet available on the company's website. Synovial tissue specimens were obtained from patients with RA and OA undergoing joint replacement surgery at the Tan Tock Seng Hospital (TTSH, Singapore). The patients were recruited from TTSH's Department of Rheumatology, Allergy and Immunology. All participants were at least 21 years of age at study entry and fulfilled the 1987 American College of Rheumatology revised criteria or the 2010 American College of Rheumatology/European League Against Rheumatism criteria for RA[56,57]. There were two RA patients (male aged above 50 years and female aged above 65) and three OA patients (two males and a female aged above 65). The sample shown in Fig. 1c originates from the right knee of an RA patient with a Disease Activity Score 28 of 4.38. All the patients were Chinese except one OA patient who was Malay. The procedures were approved by the Ethics Committee of The National Healthcare Group domain specific review board under protocol no. 2018/00980. All patients gave written consent and met the diagnostic criteria for RA and OA.

Cell line: SW982 (ATCC® HTB-93) is a synovial fibroblast cell line derived from synovium of a patient with synovial sarcoma. SW982 cells were maintained in Leibovitz's L-15 Medium (Gibco; ThermoFisher Scientific) supplemented with 10% (v/v) foetal calf serum (FCS) and 1% (w/v) penicillin/streptomycin (Gibco; ThermoFisher Scientific) at 37 °C in a free gas exchange with atmospheric air. The SW982 cells were engineered to stably express a doxycycline-inducible gene encoding the ER localised-GALNT1, wildtype GALNT1 and ER-2Lec using the Sleeping Beauty transposon system.

Mice: Col6a1Cre mice expressing collagen type VI promoter driven on the C57BL/6J background were provided by G. Bressan (University of Milano, Milano, Italy). ER-2Lec mice on the same background expressing an ER-localized double-lectin domain (ER-2Lec) under the control of a loxP-flanked STOP cassette were generated by Ozgene (Australia). Col6a1Cre ER-2Lec mice were generated by crossing Col6a1Cre mice with ER-2Lec mice, resulting in mice expressing ER-2Lec in the mesenchymal cell lineages. DBA/1J mice were obtained from the Jackson laboratory (USA) and the colony was expanded for experiments. All animals were bred and maintained under specific pathogen-free conditions in micro-isolator cages with access to food and water at Biological Resource Centre (ASTAR, Singapore). Experiments were performed using age- and sex-matched animals and complied with guidelines approved by the Animal Ethics Committees at Biological Research Centre (ASTAR, Singapore) under the protocol IACUC no. 201548.

### Isolation of primary SF from human tissues
Synovial tissues from OA and RA patients were obtained at the time of synovectomy or synovial biopsy. Following excision, the tissues were immediately minced and digested in collagenase IV (1 mg/ml, Gibco) in Dulbecco's modified Eagle medium (DMEM) for 1.5 h at 37 °C with gentle agitation. The mixture was passed through a 70 µm mesh cell strainer and cell pellet was obtained after centrifugation at 250 g for 10 min. Human SFs from OA and RA patients (OASF and RASF) were used between passages 3 and 9[58]. The purity of culture was confirmed by staining with fibroblast cell identity marker CD90 before experiments. Normal human SFs were derived from synovial tissues from a healthy human donor (HCSF) and obtained from the Cell Applications, Inc. HCSF were cultured in complete in DMEM supplemented with 10% (v/v) FCS and 1% (w/v) penicillin/streptomycin (Gibco; ThermoFisher Scientific) at 37 °C in a humidified atmosphere containing 5% CO$_2$.

### Reagents
Antibodies: Anti-CNX (ab10286, ab22595), anti-Vimentin (ab92547), anti-beta actin (ab8226), anti-GALNT2 (ab262868), anti-PDIA4 (ab82587, ab155800), anti-TGN46 (ab16059), anti-Giantin (ab37266), anti-PDIA3 (ab13506), anti-Collagen III (ab7778), anti-Fibronectin (ab2413), anti-Calreticulin (ab22683) and rabbit polyclonal isotype control (ab37415) antibodies were purchased from Abcam. Agarose bound VVL (AL-1233)) was purchased from Vector Laboratories. PE.C7 conjugated anti-CD45 (cat no. 103113), PE conjugated anti-CD90 (cat no. 328110) were purchased from Biolegend. Anti-FAPα (MAB3715) was purchased from R&D systems. Anti-NEM OX133 (Ab00579-1.1) antibody was purchased from Absolute Antibody. Anti-GALNT2 (NBP1-83394) antibody was purchased from Novus Biologicals, LLC. Anti-Collagen I (cat no. 1310-01) was purchased from Southern Biotech. Anti-TGN46 (AHP500G) was purchased from AbD Serotec. Anti-rabbit IgG-HRP (NA934) and anti-mouse IgG-HRP (NA931) antibodies were purchased from GE Healthcare Life Sciences. More information on the antibodies can be found in supplementary data 1.

Plasmids: The plasmid expressing doxycycline inducible ER-2Lec was generated as previously described[25]. The resultant vectors together with the pPGK-SB13 expressing Sleeping Beauty transposase were used to transfect SW982 cells at a ratio of 1/10 to derive stable polyclonal population.

## Immunofluorescence staining

Formalin-fixed, paraffin-embedded tissue sections were deparaffinized in xylene substitute buffer (Sub-X, Leica Biosystems) and rehydrated. For mouse joint tissues and patient tissue microarray (provitro AG, Berlin, Germany), antigen retrieval was performed by immersion in Epitope Retrieval Solution pH 6 (Leica Biosystems) and incubated at 60 °C in an oven for 18 h. For human tissue specimens, antigen retrieval was performed using Epitope Retrieval Solution pH 6 (Leica Biosystems) in a pressure chamber (2100 Retriever, Akribis Scientific Limited, WA16 0JG, GB). Sections were washed twice with PBS and immersed in a blocking buffer (5% horse serum, 1% Triton ×100). After 1 h, sections were incubated overnight at 4 °C with a primary antibody mix containing primary antibodies including rabbit anti CD45, rabbit anti Vimentin, rabbit anti-Calnexin, rat anti FAPα or biotin conjugated VVL lectin. Samples were washed thrice with blocking buffer and incubated for 2 h with a secondary antibodies including anti-rat Alexa Fluor 647-conjugated and anti-rabbit Alexa Fluor® 594-conjugated or Alexa Fluor®488 streptavidin (ThermoFisher Scientific, 1:400). After washing, cell nuclei were stained with Hoechst 33342 (ThermoFisher Scientific; 1:1000) for 5 min before mounting. All images were captured using the same settings on the LSM-700 (Zeiss) confocal microscope. Raw integrated density of VVL and nuclear signals were measured with the FIJI image calculator.

## Collagen antibody induced arthritis

On day 0, 8 weeks old male mice were injected with an antibody cocktail mix containing 5 different monoclonal antibody clones against collagen type II proteins (Chondrex Inc.) at a dose of 2 mg/mouse (200 μl) via intraperitoneal (IP) injections. On day 3, animals were injected with LPS at 50 μg/mouse via IP injections (100 μl). From day 3, arthritis severity was assessed daily by measuring paw thickness using a digital caliper and assessing qualitative clinical scores by a blinded researcher using a scoring system provided by Chondrex, Inc. For anti-CNX treatment, mice were injected with 25 μg of isotype control (ab37415) or anti-CNX antibodies (ab22595) every 2 days for a total of 4 times after LPS treatment on day 3. For anti-Calnexin scFv treatment, we used 100 μg of scFv injected every two days in mice. The control mice were injected with PBS.

## Collagen induced arthritis

DBA/1 mice were injected with an emulsion of 100 μg collagen type II and 2 mg/ml *Mycobacterium tuberculosis* (Chondrex Inc.) via subcutaneous injections. From day 7, arthritis severity was assessed every 3–5 days by measuring paw thickness using a digital caliper and assessing qualitative clinical scores by a blinded researcher using a scoring system provided by Chondrex, Inc.

## Histology analysis and Cartilage ECM staining

After removing skin, both front and hind paws were formalin-fixed and embedded in paraffin. Tissues were deparaffinized and rehydrated as described above in the above IF protocol. Tissue sections were stained with Hematoxylin and Eosin (HE), Safranin-O (SO), Alcian Blue (AB) stains. Slides were scanned at 20× using a Leica SCN400 slide scanner (Leica Microsystems, Germany). Images were exported to Slidepath Digital Image Hub (Leica Microsystems, Germany) for viewing. Selected regions were analyzed using Measure Stained Area Assay of Slidepath Tissue Image Analysis 2.0 software (Leica Microsystems, Germany). A quantitative analysis of cartilage ECM staining areas was performed using FIJI image calculator.

## Flow cytometry

Skin from the feet was removed and the joints were cut 3 mm above the heel. To avoid contamination with the bone marrow, bone marrow cavity in the tibia was thoroughly flushed with HBSS, the joints were cut into small pieces and incubated in digestion buffer (1 mg/ml collagenase IV, and 1 mg/ml of DNase I in HBSS) for 60 min at 37 °C. Cells released during the digestion were filtered through 70 μm cell strainers, erythrocytes were lysed using a red blood lysis buffer (BD Biosciences). Cells were stained with live/dead Aqua (Invitrogen) viability dyes, incubated with Fc Block at 1:50 (BD Biosciences), and stained with fluorochrome-conjugated antibodies including PE conjugated anti-CD90 (Biolegend), PECy7 conjugated anti-CD45 (Biolegend) and FITC conjugated VVL (Life Technologies) for 30 min. For intracellular staining, cells were fixed by using BD Cytofix/Cytoperm™ solution (BD Biosciences). Fixed cells were permeabilized in 1× BD Perm/Wash buffer (BD Biosciences) prior staining with FITC conjugated VVL. Samples were acquired on FACS BD LSRII and analyzed using Kaluza software (Beckman Coulter).

## Western blot and VVL-CoIP

Cells were seeded at $2 \times 10^5$ cell/ml in 10 cm dishes precoated with 2 mg/ml cartilage extracellular matrix (ECM) (Xylyx Bio.) and left to rest overnight. After stimulation with 100 μg/ml TNFα (PeproTech) and 100 μg/ml IL-1β (PeproTech) for 24 h, cells were harvested and lysed in a low-stringency RIPA lysis buffer (50 mM Tris, 200 mM NaCl, 0.5% NP-40, Complete and PhoStop inhibitor [Roche Applied Science]) and lysed for 30 min at 4 °C. Lysates were then clarified by centrifugation at 13,000 × g for 10 min at 4 °C. Clarified tissue lysates were incubated with agarose-bound VVL beads (Vector Laboratories) overnight at 4 °C. Beads were washed 3 times with RIPA lysis buffer, and the precipitated proteins were eluted in 2× SDS sample buffer containing 50 mM DTT. Lysates were boiled at 95 °C for 5 min and separated by SDS-PAGE electrophoresis using 4–12% Bis-Tris 80 NuPage gels (Invitrogen) at 180 V for 70 min. Samples were then transferred on nitrocellulose membranes using iBlot transfer system (Invitrogen) and blocked using 3% BSA dissolved in TBST (50 mM Tris, 150 mM NaCl and 0.1% Tween-20) for 1 h at room temperature. Membranes were then incubated with primary antibodies (1/1000 diluted in 3% BSA-TBST) overnight at 4 °C. The next day, membranes were washed 3 times with TBST and incubated with secondary antibodies conjugated with horseradish peroxidase (HRP) for 2 h at room temperature. Membranes were washed 3 more times with TBST before ECL exposure.

## Matrix degradation assay

Red gelatin coverslips were prepared as previously described[29]. The coverslips were coated with 0.2 mg/ml cartilage ECM (Xylyx Bio) for 3 hr at 37 °C. SFs cells (including SW982 cells, OASF, RASF and HCSF) were seeded at 50,000 cells/ml/well in a 24-well plate overnight. Cells were then stimulated with 100 ng/ml TNFα and 100 ng/ml IL-1β. After 24 h, the cells were fixed with 4% PFA and stained for nucleus using Hoechst 33342 (Life Technologies). For the experiment to test the effect of anti-CNX, OASF cells were incubated with 5 μg of anti-CNX antibody (ab22595, abcam) or the isotype control for 48 h before fixation. Stained coverslips were mounted onto glass microscope slides and 10 to 30 images were acquired for each condition. The normalized area of matrix degradation relative to the number of cells was measured by using ImageJ software (version 1.53t) as previously described by Martin et al.[59]. Briefly, the area of degradation using the fluorescent gelatin images after thresholding and the same threshold were applied for all images. The cell counter tool was used to count the number of nuclei and the area of gelatin degradation per total number of cells was calculated. Experiments were done in 3 biological replicates.

**Statistical analyses**

GraphPad Prism (version 8.4.3, GraphPad Software, CA, USA) was used for statistical analyses and graphical preparation. Data were analyzed by one-way (Kruskal Wallis test), two-way ANOVA (Tukey's multiple comparisons test) or Mann–Whitney test as indicated. Differences were considered statistically significant for $p$ values < 0.05.

**Reporting summary**

Further information on research design is available in the Nature Portfolio Reporting Summary linked to this article.

## Data availability

All data supporting the findings described in this manuscript are available in the article and in the Supplementary Information and from the corresponding author upon request. Source data are provided with this paper.

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

## Acknowledgements
The human tissue collection for this study was supported by Centre Grants (CG12Aug17 and CGAug16M012) and an individual research grant (NMRC/CG/017/2013) awarded by the National Medical Research Council, Ministry of Health, Singapore to Tan Tock Seng Hospital. FB is supported by grants from AMIDEX (AMX-20-CE-03) and from the Fondation ARC pour la recherche sur le cancer. J.C. is supported by A*STAR Career development fund (2021; 212D800073). We thank Dr Tong Leng Tan and Dr Sreedharan S/O Sechachalam, Department of Orthopaedic Surgery, Tan Tock Seng Hospital, for harvesting the samples during surgery. We thank Ms Ying Qi Choong, Ms Mei Yun Yong, Ms Chia Mun Woo, Mr Kia Wei Chua, Ms Amelia Lim, Ms Jocelyn Huiyun Gay and Ms Joo Yong Ong for data and sample collection.

## Author contributions
L.S.T. and J.C. designed and performed most experiments, X.L.G. generated the scFV against CNX and related experiments, T.K.M. generated the CIA mouse model and performed the animal work, A.T.N. and W.C.C. contributed in some animal experiments, V.S. contributed to the study design. T.T.L. and S.S. performed surgery and harvested patient samples, L.K.P. contributed to the discussions and manuscript writing, recruited patients and coordinated patient sample collection, F.B. designed experiments and wrote the manuscript.

## Competing interests
There is potential Competing Interest. F.B., J.C., X.L.G., and A.T.N. have active participation with Albatroz therapeutics Pte Ltd. A Singapore Patent Application no: 10202100687X has been filed in relation to the work presented in this study. The other authors declare no competing interests.
