## [Transparent Peer Review file · Nature Communications]

ER O-glycosylation in synovial fibroblasts drives cartilage degradation

Corresponding Author: Dr Frederic Bard

A version of this paper was originally rejected for publication by Nature Communications, however that decision was reconsidered after appeal by the authors.

Version 0:

Reviewer comments:

Reviewer #1

(Remarks to the Author)

The authors present an interesting study on the potential of inhibiting calnexin-mediated ECM degradation to treat RA. There is very little details and methods regarding the studies with the anti-CNX antibodies. The authors need to make significant updates to clarify and fully describe these areas before the manuscript can be considered for publication.

- 1) What isotopes are these Calnexin antibodies? Is effector function anticipated to play a role in function?
- 2) The authors should provide more details on in vitro ECM degradation and in vivo studies with the anti-CNX antibodies. Specifically, which antibodies, concentrations, and controls were used?
- 3) What polyclonal antibodies were used in the in vitro ECM degradation assay? Why were monoclonal antibodies not used? Related to this, if only polyclonal antibodies were used, this does not necessarily mean that a monoclonal IgG will have therapeutic potential. The authors should mention and discuss this caveat.
- 4) What is the mechanism of antibody inhibition of calnexin?

Reviewer #2

(Remarks to the Author)

In this paper, Tran and colleagues report their findings on O-glycosylation of surface proteins in synovial fibroblasts. They demonstrate that increased O-glycosylation in RA and OA synovial tissues is associated with polypeptide N-acetylgalactosaminyltransferases (GALNTs) activation. Based on a variety of in vitro and in vivo data (largely obtained in the CIAI model) they suggest that modulating O-glycosylation of surface proteins by GALNTs inhibitors or anti-Calnexin antibodies may prevent cartilage degradation and constitute a more therapeutic approach.

Overall, this is an interesting and innovative concept that is based on previous observations in cancer cells and that may indeed stimulate further research in this not very well-developed field. Unfortunately, there are a number of weaknesses particularly with respect to the translational value (and thus the main conclusion) of the observations:

- The human data, overall, are quite weak and largely based on some (immune)staining. The lack of a difference between RA and OA FLS along with the high variability may suggest that the observed changes are transient and unspecific. The authors suggest that "... GALA is only fully activated during the active ECM degradative phase of the disease, a phase that would correspond to flares in patients. By contrast, in phases of remission, there would be lower low glycosylation levels and correspondingly less ECM degradation" but this is pure speculation and not supported by any data. In order to substantiate the underlying hypothesis it is essential that the authors define far more precisely in which patient populations and under which circumstances there's measurable changes on the O-glycosylation of surface proteins in synovial fibroblasts. (e.g., is this related to disease activity, radiological progression, subtypes of disease etc.)

- similarly, an in-depth characterization of the FLS is missing. Key questions include that of differences between RA and OA FLS (e.g. with respect to the affected proteins and the durability of the changes in vitro). This is of particular importance as RS-FLS have been described to show lasting alterations in key characteristics (sometimes called 'tumour-like

transformation') and an important question of course is whether increased O-glycosylation of surface proteins belongs to these lasting changes.

- the authors have chosen a quite acute animal model and it seems to be essential to test their hypothesis in a second, more chronic model that better reflects key mechanisms of human RA. This is because the CIAI model actually bypasses important mechanisms of human RA, particularly with respect to FLS biology

Reviewer #3

(Remarks to the Author)

In this manuscript, the authors propose that the relocation of Golgi-resident GALNTs to the ER results in the O-glycosylation of the ER protein calnexin, which then leads to calnexin being exported to the cell surface, where it participates in matrix degradation. This is an interesting model but there are some concerns regarding the use of Calnexin (the protein proposed to be relocated) as the marker to show ER localization. Additional points to be clarified are below:

1. Can the authors explain what the ER-2Lec is and how it inhibits GALNTs?
2. Is there evidence in patient synovial tissues that GALNT2 localizes to the Golgi in normal tissues and localizes to the ER in diseased states (using both Golgi and ER markers along with GALNT2 Abs)?
3. Is GALNT2 expression upregulated in diseased tissues? Is Calnexin expression upregulated in diseased tissues?
4. Why was GALNT2 chosen to examine—is GALNT2 the major enzyme in these tissues?
5. If the model is that O-glycosylation of calnexin causes it to localize to the cell surface, then calnexin should not be used as an ER marker in this study.

Fig.1a and e—While there appears to be an increase in VVL between normal vs. diseased tissue, it would be informative to know whether this is specific to VVL. Is there also an increase in the staining of other lectins, such as those that detect extended O-glycans or N-glycans? There may in fact be many differences between normal and diseased tissues (particularly if ER stress is involved in response to the local inflammation).

Fig. 1g—Given that the authors propose that Calnexin is O-glycosylated by GALNT2, VVL would then be recognizing Calnexin (indeed, some overlap of VVL and Calnexin staining can be seen in the UT sample). Therefore, using Calnexin as a marker for the ER to show VVL ER localization becomes problematic. Moreover, this study also posits that O-glycosylation of calnexin causes it to move to the cell surface. Therefore a different marker for the ER should be used.

Fig. 1h—Given the resolution of these images and the fact that a Golgi marker has not been included, it is impossible to conclude that GALNT2 has moved from the Golgi to the ER. In fact, it looks like there has been a dramatic increase in the expression of GALNT2 and a dramatic expansion of calnexin staining (which because of its putative relocation to the cell surface may not just be staining the ER), potentially indicative of ER stress.

Fig. 2d—This experiment is used to support a role for cytokines in increasing matrix degradation in subsequent experiments (in Fig. 3). However, the groups that are being compared do not address whether or not cytokines increase HPL intensity in OASF or RASF cells (e.g. OASF untreated vs OASF cyto, etc....).

Fig. 2e—This is very difficult to distinguish the color for HPL and Giantin here. Again, both ER and Golgi markers should be included to quantitate the degree of overlap with each compartment.

Fig. 4—Is the increase in Calnexin seen via FACS due to increased expression of Calnexin or a change in the cell surface localization? Do the authors have high magnification views of untreated and treated cells stained for calnexin, another ER marker and a cell surface marker to assess changes in calnexin localization?

Version 1:

Reviewer comments:

Reviewer #1

(Remarks to the Author)

The authors have addressed my comments.

Reviewer #3

(Remarks to the Author)

The authors state the following as the major conclusions of their paper (last paragraph in the Introduction) which are largely unsupported by direct data. This paper therefore does not meet the standards of publication for Nature Communications:

In this report, we show that lining SFs from arthritic patients but not from healthy individuals display marked ER relocation of GALNTs. The resulting increase of O-glycosylation, driven by re-compartmentation of the enzymes, is critical for disease progression in mice. The ER-resident

protein Calnexin gets glycosylated and translocated to the cell surface where it participates in ECM degradation. We show that inhibiting Calnexin directly alleviates RA symptoms. In sum, we show that regulation of protein glycosylation by this non-transcriptional process is associated with the active phase of arthritis and can be exploited therapeutically.

The authors have failed to demonstrate that the GALNTs are relocated to the ER in patient samples. The authors state that the GALNT2 antibody did not work in patient samples. Therefore, the conclusion that GALNT2 relocalizes to the ER in arthritic patients cannot be made. As the authors are aware, lectin staining (lectins recognize glycans attached to proteins) cannot be used as a surrogate for the GALNT enzymes as many glycosylated proteins are known to move throughout the cell via anterograde transport, retrograde transport, endocytosis, etc.....after being glycosylated. VVL/HPL staining are detecting (likely) hundreds of different proteins that have had a GalNAc added to them and may be retrotransported to the ER. This has no bearing on where the enzymes that glycosylate them are located.

Additionally, in instances where the authors looked at GALNTs directly in mouse tissues, costaining with both a Golgi and an ER marker should be done to conclusively demonstrate that the GALNTs have moved from the Golgi to the ER. While they did include an ER marker, it is again one that is purported to form a complex with the hyperglycosylated calnexin (PDIA3) in the experimental state and cannot be used as a true ER marker.

In the instance where they looked at GALNT directly in cells with a Golgi and ER marker, it appears that most of the GALNT staining is still in the Golgi (Fig. 2f). So this is clearly not as simple as “re-compartmentation” of the GALNTs to the ER, as most staining remains in the Golgi.

The second conclusion is that glycosylated calnexin is translocated to the surface of the cells and is responsible for the ensuing ECM degradation and disease progression is not adequately supported with direct data. Is the cell surface calnexin specifically hyperglycosylated as their model predicts? With stimulation, what percentage of calnexin is hyperglycosylated and moved to the cell surface? In their rebuttal, the authors state that calnexin can be used as a marker for the ER as most of the calnexin remains in the ER even under stimulated conditions. What does that mean in terms of this model?

Finally, this paper implies that, based on the unsupported conclusions above, a therapeutic option for OA and RA patients may be antibodies to calnexin. Given these unknowns, and the unknown mechanism by which the ECM is actually being degraded, it is difficult to interpret what antibodies to calnexin might be doing in vivo. Are these antibodies specific to the hyperglycosylated form of calnexin? What effect might they be having on the regular functions of calnexin? Are they influencing secretion or protein folding of other factors, such as proteases? Are they influencing immune cells? Given the published roles for calnexin in respiration and T cell migration, intraperitoneally injected calnexin antibodies could be having a plethora of effects on immune system function unrelated to the proposed mechanism here. We should be mindful not to overstate the potential therapeutic relevance without a thorough investigation of the mechanism. Indeed, OA and RA are very complex and distinct diseases, with unique pathologies, immune system involvement, risk factors and progressions. There are currently no data to indicate that any of the GALNTs undergo “re-compartmentation” in either OA or RA patients.

Reviewer #4

(Remarks to the Author)

The authors have performed substantial work to extend and support their study, however, extrapolation from the human immune staining still overreaches the data presented and should be discussed in a more balanced way.

‘Human tissue microarray (Catalog number 401 3201) of synovial tissues was obtained from provitro AG (Berlin, Germany). Patients’ information were provided in the datasheet available on the company’s website’.

The datasheet on the company’s website shows only limited data from 28 tissue spots – not the complete data from all 52 samples assessed. The clinical data for all of the tissue stained must be included as a supplementary table. Information including age, tissue biopsy location, gender, disease duration, DAS, ESR/CRP, tissue erosion, and treatment would be useful. Does GALA activation correlate with any of these metadata? For example if there is no association with markers of inflammation but with the numbers of erosions this would support a link between increased O-glycosylation and tissue destruction, rather than inflammation.

“In patient samples, Tn is detectable in all samples of OA, RA and psoriasis arthritis, but the levels are variable. This could be explained if GALA is only fully activated during the active ECM degradative phase of the disease, a phase that would correspond to flares in patients. By contrast, in phases of remission, there would be less ECM degradation and correspondingly lower glycosylation levels”.

Whilst it is true that the heterogeneity in Tn levels could be due to disease in flare or remission there are no human data to support this in this manuscript and it is difficult to make firm conclusions about human disease from the CAIA model. It is also true that the differences observed could be due to any number of other clinical features including but not limited to disease activity, disease duration and/or treatment response. The groups of OA and RA examined here are relatively small and a

much larger study with better defined patient cohorts and more detailed clinical metadata are needed to be able to support any conclusion about human disease staging. This lies beyond the current study but the conclusions on page 14 must be amended and a more balanced discussion added. The need for better stratification of patients in much larger independent cohorts should also be discussed.

Version 2:

Reviewer comments:

Reviewer #3

(Remarks to the Author)

It appears as though the authors have replied to my previous critique by citing previously published studies from their group, rather than performing additional analyses or presenting new data. The conclusions of this paper should be based on the data within this paper, not on previously published studies in other systems. Given that there appears to be no new data or experiments, I will reiterate my primary concerns.

The authors have failed to demonstrate that the GALNTs are relocated to the ER in patient samples. The authors state that the GALNT2 antibody did not work in patient samples. Therefore, the conclusion that GALNT2 relocalizes to the ER in arthritic patients cannot be made.

Additionally, in instances where the authors looked at GALNTs directly in mouse tissues with an ER marker, they used Calnexin as the ER marker, which is supposed to be the protein that is mis-located to the cell surface in this model. The authors respond that the GALNT relocation to the ER is not "on-off" but is variable, with most of the GALNTs remaining in the Golgi. Likewise, they respond that the majority of Calnexin remains in the ER, so it can be used as an ER marker. This raises concerns as to how this model can be rigorously tested if it is unclear to what extent mislocation occurs and how this does or does not correlate with phenotypes observed. How much GALNT in the ER is needed to cause the effects? How much Calnexin at the cell surface is abnormal? What percentage of relocation of the GALNTs and Calnexin is necessary to see the phenotypes? Is all calnexin within the ER glycosylated normally and all on the cell surface abnormally glycosylated?

Given the lack of data for GALNT relocation in patient samples and the lack of correlation between Tn levels and synovitis score, it would be premature to suggest antibodies to calnexin as a potential therapy.

Reviewer #4

(Remarks to the Author)

The reviewers have addressed my comments satisfactorily

Version 3:

Reviewer comments:

Reviewer #5

(Remarks to the Author)

In this study, the authors used human samples, mouse models, and different cell lines to test the role of O-glycosylation in ECM remodeling in arthritic diseases. They provided evidence that Golgi-localized GALNTs are relocated to the ER in activated arthritic synovial fibroblasts (SFs), leading to increased glycosylation and cell surface localization of the ER chaperone Calnexin. Calnexin participates in matrix degradation by reducing ECM disulfide bonds; anti-Calnexin antibodies block ECM degradation and protect animals from arthritic diseases. Altogether, the authors conclude that ER O-glycosylation in synovial fibroblasts drives cartilage degradation. Overall, the topic is interesting, it reveals a novel role of O-glycosylation in ECM remodeling in arthritic diseases.

Major concerns:

1. There is a lack of solid evidence to support the ER localization of GALNTs in activated arthritic synovial fibroblasts.

Figure 1ab (and following figures), using the VVL signal to represent the localization of GALNT is indirect, as many GALNT substrates are supposed to be transported from the ER to other cellular locations. In addition, the VVL signal in RA and OA samples is in average 3-fold of control in Figure 1b, not seven-fold as the authors claimed.

Supplementary Fig. 1c, the GFP signal in the Golgi-G1 cells is mostly in the ER, not Golgi. It is unclear why the HPL signal is not increased in Golgi-G1 cells as in the ER-G1 cells regardless of the localization of the enzyme.

Supplementary Fig. 2d, while the authors showed that the GALNT2 signal in synovial fibroblasts at the forefront exhibited diffused, ER-like staining, but they did not provide the result in control tissues/cells for comparison.

Supplementary Fig. 3a, it seems that the HPL signal is mostly in the Golgi in almost all cells. GALNT localization is not shown in this figure.

Figure 2f, depending on the experimental procedure (such as the blocking method and the concentration of the antibody used) and the background setting on the microscopy, this result may vary. This important result should be validated by other methods.

There is also no mechanism of how GALNTs are relocalized to the ER under disease conditions.

2. No evidence is provided to support that GALNT localization to the ER is the cause of elevated VVL level or Calnexin trafficking to the plasma membrane in disease models.

The expression level of GALNTs appears to vary in different results; evidence is needed to confirm that it is GALNT ER localization, rather than increased expression, is the cause of the observed effects. For example, in Figure 1h, while it is true that the signal in the control appear as puncta-like structures, the signal in the disease model is much stronger, raising a concern of whether the appearing broader localization is due to increased expression. Higher magnification images are needed with similar signal intensities to confirm the localization of the proteins. Adding a Golgi marker may also be helpful in this study. In addition, can the authors determine the expression level of GALNT2 by western blotting in this and other experiments?

How could Calnexin glycosylation in the ER lumen affects its ER retention signal at the cytoplasmic tail? Calnexin cleavage has been reported in the literature, have the authors consider this possibility?

Many experiments are incomplete, the authors used different systems to show different results, such as VVL signal level in tissues (Fig. 2ab) and GALNT localization in cells (Fig. 2fg), making it difficult to correlate the different results in the same experimental setting.

3. There is no mechanism/evidence of how cell surface location of Calnexin affects ECM degradation. The authors reasoned this as the secretion of PDIA3 but did not provide experimental evidence.

Figure 3, there is no result shown to correlate the expression level/localization of exogenous and endogenous GALNT1 with ECM degradation.

Figure 4, GALNT1 blot should be shown. Actin blot shows that the loading is uneven, making the result inconclusive.

Supplementary Fig. 5b, since these are permeabilized cells, why there is no signal in control cells? This raises a concern that increased GALNT expression is more important than its ER-localization.

Here is a speculation of how adding anti-calnexin antibodies to cells or animals could reduce ECM degradation: calnexin is a lectin that binds glycoproteins including MMPs, and antibody binding would trigger endocytosis and degradation of cell surface calnexin together with MMPs, leading to the reduction of ECM degradation.

4. The writing needs to be improved. Examples include "autoreactive B and T cells produce autoantibodies" (T cells do not produce antibodies), "the relocation of GALNTs to the ER is a reversible event under the control of signaling pathways" (this is not shown in the paper), and "2x LDS sample buffer" (might be SDS sample buffer?). In addition, abbreviations should be spelled out when appear first time.

Version 4:

Reviewer comments:

Reviewer #5

(Remarks to the Author)

The authors have addressed my critiques, this reviewer has no further questions.

Re-compartmentation of O-glycosylation in synovial fibroblasts drives cartilage degradation

Letter in response to the reviewers

Dear Editor and reviewers,

We thank you for the careful reading of our manuscript. We have spent quite some time developing the tools and methods required to answer some of the questions and addressing concerns. For instance, we sourced and bred a colony of a special strain of mice (DBA/1J) to set-up an auto-immune arthritis assay (collagen type II induced). We have screened a library and isolated an antibody fragment (ScFv) against Calnexin in order to verify our initial findings. We hope you will find the large body of work that extends the initial findings significant. In the manuscript, we have put in blue the text corresponding to new data.

We appreciate that it has been a while since our initial submission and are grateful for your attention.

Points of discussion

Reviewer #1:

1) What isotopes are these Calnexin antibodies? Is effector function anticipated to play a role in function?

The isotope used in the original version was IgG from rabbit polyclonal. We cannot exclude that the effector function may play a role but we believe that blocking the surface function of Calnexin is most critical. In an effort to develop the translational potential of targeting Calnexin, we screened an scFv library against the luminal domain of Calnexin. We next injected mice with the scFv and observed a reduction of arthritic symptoms. This data confirms that targeting Calnexin can have an effect on arthritis and does not require any effector function (See Supplementary Fig. 5).

Text added:

“To validate our findings, we employed a different antibody format, a single chain Fv (scFv), which specifically targeted CNX and was developed in our lab. The specificity of the anti-CN X scFv was confirmed through western blotting (Supplementary Fig. 5a) and immunofluorescence using CNX-mCherry expressing cells (Supplementary Fig. 5b). Subsequently, CAIA mice were treated with the scFv by injecting 100 micrograms every two days from day 3 to day 9 after CAIA initiation (Supplementary Fig. 5c). Similar to the results observed with anti-CN X IgG1 (Figure 5f), significant relief of arthritis symptoms was observed in animals treated with anti-CN X ScFv compared to the control group Supplementary Fig. 5d, e). Histological stainings further revealed detectable scFv signal in the synovial lining of the joints, co-localizing with the fibroblast marker vimentin (Supplementary Fig. 5f).”

Supplementary Fig. 5f:

Supplementary Fig. 5: Blockage of CNX with single chain Fv (scFV) protects CAIA mice from arthritis.

(a) Western Blot analysis of CNX with identified scFv against CNX. Cell lysates from MDA-231 and MDA-231 CNX^{-/-} probed with myc tagged scFv specific for CNX and subsequently with anti-myc Horseradish peroxidase. A loading control with tubulin is indicated. (b) Representative Immunofluorescence pictures with scFv against CNX. Fixed and permeabilized Huh7 normal cells or Huh7 cells with stably integrated mcherry CNX (orange) subjected to incubation with myc tagged scFv and subsequently to secondary anti-Myc conjugated to PhycoErythrin (green) and Hoechst nuclear dye (blue). Representative pictures for CNX mcherry stable cells are indicated on top (left) along with scFv co-stain (right). Representative pictures for Huh7 control cells are indicated on the bottom (left) along with scFv co-stain (right). All images are from constant acquisition and display settings. scale bar 20µm.

(c) Experimental setup of injections and timeline used for treatment of collagen antibody induced arthritis mouse model with scFv against CNX.

(d) Paw thickness variation in CAIA mice treated with scFv injection or control PBS injection from day 0 to

day 10.

(e) Clinical scores of CAIA mice treated with scFv injection or control PBS injection from day 0 to day 10.

(f) Representative immunofluorescence images of synovial tissues sections from CAIA mice day 10 from control PBS or myc tagged scFv injected mice stained with anti-Myc conjugated to PhycoErythrin (green), Vimentin and DAPI. Scale bar, 40 μ m.

2) The authors should provide more details on in vitro ECM degradation and in vivo studies with the anti-CNX antibodies. Specifically, which antibodies, concentrations, and controls were used?

We have expanded the text in material and methods and in figure legends.

Material and Methods:

“Collagen antibody induced arthritis

On day 0, 8 weeks old male mice were injected with an antibody cocktail mix containing 5 different monoclonal antibody clones against collagen type II proteins (Chondrex Inc.) at a dose of 2 mg/mouse (200 μ l) via intraperitoneal (IP) injections. On day 3, animals were injected with LPS at 50 μ g/mouse via IP injections (100 μ l). From day 3, arthritis severity was assessed daily by measuring paw thickness using a digital calliper and assessing qualitative clinical scores by a blinded researcher using a scoring system provided by Chondrex, Inc. For anti-CNX treatment, mice were injected with 25 μ g of isotype control (ab37415) or anti-CNX antibodies (ab22595) every 2 days for a total of 4 times after LPS treatment on day 3. For anti-Calnexin scFv treatment, we used 100 μ g of scFv injected every two days in mice. The control mice were injected with PBS.

Matrix degradation assay

Red gelatin coverslips were prepared as previously described³⁰. The coverslips were coated with 0.2 mg/ml cartilage ECM (Xylyx Bio) for 3 hr at 37°C. Synovial fibroblasts cells (including SW982 cells, OASF, RASF and HCSF) were seeded at 50,000 cells/ml/well in a 24-well plate overnight. Cells were then stimulated with 100 ng/ml TNF α and 100 ng/ml IL-1 β . After 24 hr, the cells were fixed with 4% PFA and stained for nucleus using Hoechst 33342 (Life Technologies). For the experiment to test the effect of anti-CNX, OASF cells were incubated with 5 μ g of anti-CNX antibody (ab22595, abcam) or the isotype control for 48h before fixation. Stained coverslips were mounted onto glass microscope slides and 10 to 30 images were acquired for each condition. The normalized area of matrix degradation relative to the number of cells was measured by using imageJ software as previously described by Martin *et al*⁵⁸. Briefly, the area of degradation using the fluorescent gelatin images after thresholding and the same threshold were applied for all images. The cell counter tool was used to count the number of nuclei and the area of gelatin degradation per total number of cells was calculated. Experiments were done in 3 biological replicates.

Figure legends:

Fig 5: Blockage of CNX protects CAIA mice from cartilage degradation and development of severe arthritis

(b & c) Representative images (b) and quantification graph (c) show decreased matrix degradation activity in primary fibroblasts isolated from OA patients after 48h incubation with 5 μ g anti-CNX antibodies (ab22595, abcam) as compared to those treated with isotype control antibodies (ab37415, abcam). Data correspond to the mean \pm SEM and representative data for three independent experiments. Each data point represents the total degraded area (μ m) per nuclear per well. ***, p<0.001 (Mann-Whitney test).

(d) Antibody treatment scheme. Mice were intraperitoneally injected with 2mg anti-collagen II antibodies (Chondrex Inc.) on day 0, followed by 50 μ g lipopolysaccharide (LPS) injection on day 3. Antibody treatment was subsequently administered at 25 μ g of antibody per mice on the same day and the treatment was repeated every 2 days. A total of 4 antibody treatments was given for each treated mouse.”

3) What polyclonal antibodies were used in the in vitro ECM degradation assay? Why were monoclonal antibodies not used? Related to this, if only polyclonal antibodies were used, this does

not necessarily mean that a monoclonal IgG will have therapeutic potential. The authors should mention and discuss this caveat.

As noted above, we have a rabbit polyclonal antibody (ab22595). We have also treated the mice with a monoclonal scFv which alleviated disease severity (Supplementary Fig. 5).

We agree with the reviewer and have pointed out the current limitations of the approach in the discussion:

“The role of Calnexin in the degradation of ECM was only recently established. In complex with PDIA3, Calnexin participates in the reduction of disulfide bonds in liver ECM proteins³⁰. Disulfide bonds, like other cross-linking bonds, prevent the action of proteases⁴⁸. The cartilage ECM contains an abundance of disulfide bonds. Anti-Calnexin antibodies blocked matrix degradation by SFs in vitro. This approach provided a significant protection of cartilage ECM and arthritic symptoms in animals. **It is to be noted however that the approach is still in its infancy. We have only used a polyclonal rabbit antibody and a fragment scFV antibody. Furthermore, the animals were treated at early time points in the development of the symptoms; it is not clear what effect a monoclonal antibody would have on a full blown arthritic crisis for example.**”

4) What is the mechanism of antibody inhibition of calnexin?

We recently reported that Calnexin plays a key role in ECM degradation by mediating disulfide bonds reduction. We believe the antibodies are preventing Calnexin from doing this job of reducing cross-links in the ECM.

To verify this mode of action, we quantified the amount of reduced cysteines in the ECM fibers after incubation with SF cells.

Results:

“Anti-CNX antibodies prevent arthritic symptoms in CAIA mice

To examine the antibody's ability to block disulfide bond reduction, SW982 ER-G1 cells were sparsely seeded on cartilage ECM and treated with either isotype or anti-CNX antibody for 2 days. The preparation was then treated with NEM to detect reduced cysteine residues using the OX133 antibody³⁰. Control cells showed strong OX133 signal around collagen I fibers (Supplementary Fig. 4a). In contrast, the presence of anti-CNX antibody significantly reduced OX133 labeling (Supplementary Fig. 4a). Quantification revealed approximately a 60% reduction in the intensity of OX133 signal in collagen I fibers surrounding the cells with anti-CNX antibody (Supplementary Fig. 4b,c).”

Reviewer #2:

- 1) - The human data, overall, are quite weak and largely based on some (immune)staining. The lack of a difference between RA and OA FLS along with the high variability may suggest that the observed changes are transient and unspecific. The authors suggest that "... GALA is only fully activated during the active ECM degradative phase of the disease, a phase that would correspond to flares in patients. By contrast, in phases of remission, there would be lower low glycosylation levels and correspondingly less ECM degradation" but this is pure speculation and not supported by any data. In order to substantiate the underlying hypothesis It is essential that the authors define far more precisely in which patient populations and under which circumstances there's measurable changes on the O-glycosylation of surface proteins in synovial fibroblasts. (e.g., is this related to disease activity, radiological progression, subtypes of disease etc.)**

We disagree with the assessment that the human data is weak. We have run an analysis on more than 52 different patient samples and we have analyzed live primary synovial fibroblast cells obtained from patients. The biopsy data shows very clearly the increase in glycosylation. The patients' cells show a clear tendency to activate the pathway more strongly.

The lack of difference between OA and RA is not a concern in our view, quite the opposite. It shows that hyperglycosylation is a mechanism common to the two diseases. There are lots of similarities between the two diseases and in particular the fact that they both imply cartilage ECM degradation.

Our remark about the fact that the pathway is probably not constantly activated in patients is based on data obtained in the mouse model, which shows that the high glycosylation status resorbs when the arthritic flare starts to resorb.

The full quote from our text was:

Whether activated by immune signals or by ECM proteins, increased O-glycosylation may not be continuous during arthritis. Indeed, in the CAIA mouse model, Tn levels significantly decreased at day 10, preceded by the full recovery of the animals (at day 14 or later). In patient samples, high Tn is detectable in samples of OA, RA and psoriasis arthritis, but a significant fraction of samples display lower to normal Tn levels. This suggests that GALA is only fully activated during the active ECM degradative phase of the disease, a phase that would correspond to flares in patients. By contrast, in phases of remission, there would be lower low glycosylation levels and correspondingly less ECM degradation.

We have modified it to:

"In patient samples, Tn is detectable in all samples of OA, RA and psoriasis arthritis, but the levels are variable. This could be explained if GALA is only fully activated during the active ECM degradative phase of the disease, a phase that would correspond to flares in patients. By contrast, in phases of remission, there would be less ECM degradation and correspondingly lower glycosylation levels."

- 2) - similarly, an in-depth characterization of the FLS is missing. Key questions include that of differences between RA and OA FLS (e.g. with respect to the affected proteins and the durability of the changes in vitro). This is of particular importance as RS-FLS have been described to show lasting alterations in key characteristics (sometimes called 'tumour-like transformation') and an important question of course is whether increased O-glycosylation of surface proteins belongs to these lasting changes.**

We have added an in depth characterisation of the FLS. We found that both OA and RA FLS have an increased capacity to activate the high glycosylation, rather than constantly high glycosylation levels. This is clearly observable in Fig 2d and e. The lasting changes mentioned by the reviewer would then be in the capacity to quickly and strongly activate the pathway. One difference that we observed between OA and RA FLS is that RA FLS activate the pathway more readily in response to cytokines, while OA FLS respond mostly to exposure to ECM. However, we would like to point out that identifying differences between the two cell types (OA and RA FLS) is not the focus of this study.

- 3) the authors have chosen a quite acute animal model and it seems to be essential to test their hypothesis in a second, more chronic model that better reflects key mechanisms of human RA. This is because the CIAI model actually bypasses important mechanisms of human RA, particularly with respect to FLS biology**

We agree that it is important to check other arthritis models. At the same time, it is also important to recognize that degradation of cartilage is the ultimate process that leads to loss of function in both forms of arthritis. It happens downstream of the immune system deregulation occurring in RA. Specifically, we are not claiming that activation of GALA in FLS has something to do with the dysregulation of the immune system occurring in human RA.

Nonetheless, we have set-up in the lab a mouse model of collagen induced arthritis.

We added this part in the results section:

"To test the change in levels of glycosylation in another model of arthritis, we used collagen induced arthritis (CIA) model. DBA strain mice were induced by immunization with type II collagen injections emulsified with

adjuvant³². These mice took about 21 days to exhibit the first signs of arthritic swelling with more severe swelling at 28-33 days. The symptoms persisted up to 35 days post induction (Supplementary Fig. 1h, i). We observed pannus invasion into the joint cavity at day 28 post injection (Supplementary Fig.1j) and reduced cartilage in the joint as revealed by Safranin O (SO) staining (Supplementary Fig. 1k). At day 35, the cartilage was almost completely excised with significant hyperplasia of cells in the joint area. High Tn levels could be seen in the cells of the invading pannus at days 28 and 35 (Supplementary Fig.1l). These results confirm that enhanced O-glycosylation is occurring in the cells of the arthritic joints and could play a role in the disease progression.”

Reviewer #3:

1. Can the authors explain what the ER-2Lec is and how it inhibits GALNTs?

We added the following text in the results section:

“Current small molecule inhibitors of GALNTs do not allow for ER-specific inhibition of GALNTs. To achieve this goal, we turned to the ER-specific GALNTs inhibitor ER-2Lec, which we described previously (Gill et al. 2013). Briefly, this inhibitor is derived from the lectin domain of GALNT2, which binds Tn. The GALNTs lectin domain mediates clustered glycosylation (multiple adjacent Thr or Ser residues modified); ER-2Lec is thought to inhibit this process and does reduce Tn levels in GALA positive cells.”

2. Is there evidence in patient synovial tissues that GALNT2 localizes to the Golgi in normal tissues and localizes to the ER in diseased states (using both Golgi and ER markers along with GALNT2 Abs)?

We found that GALNT2 is more localised in the ER in arthritic synovial membrane cells than in control tissue in mouse tissue (Fig 1h).

Unfortunately, the antibodies against GALNT2 did not work very well on the human samples available to us. We have conducted instead a VVL/TGN46 staining on a RA patient sample from the Tan Tock Seng Hospital (Supplementary Fig. 2c). This staining illustrates clearly a shift in distribution of Tn (recognized by the lectin VVL) from being Golgi localised (colocalizing with TGN46) to a cytoplasmic distribution.

In addition, in vitro, stimulation of RA FLS with cytokines induces a clear relocation of GALNT activity to the ER (Figure 2e).

Finally, we show that overexpression of GALNT2 does not result in increased Tn levels in vitro, while overexpression of ER-localised GALNT2 does so very clearly (Supplementary Fig. 1c,d,e).

3. Is GALNT2 expression upregulated in diseased tissues? Is Calnexin expression upregulated in diseased tissues?

There is indeed some upregulation of GALNT2 protein in diseased tissues (Supplementary Fig. 1f). We have also seen evidence at the mRNA level of limited upregulation (not shown). We have also found some evidence of upregulation of Calnexin, but the amplitude is also not very important (not shown).

4. Why was GALNT2 chosen to examine—is GALNT2 the major enzyme in these tissues?

Indeed, GALNT2 is one of the major GALNT enzyme expressed in FLS and in joint based on databases. GALNT1 and 2 are the most ubiquitously expressed GALNTs enzymes in tissues.

5. If the model is that O-glycosylation of calnexin causes it to localize to the cell surface, then calnexin should not be used as an ER marker in this study.

To be clear, the reason we were using Calnexin is because only a limited fraction of the protein is translocated to the cell surface. Most of Calnexin staining remains at the ER. However, to be more rigorous and facilitate the reader's comprehension, we have repeated the staining with PDIA3.

Fig.1a and e—While there appears to be an increase in VVL between normal vs. diseased tissue, it would be informative to know whether this is specific to VVL. Is there also an increase in the staining of other lectins, such as those that detect extended O-glycans or N-glycans? There may in fact be many differences between normal and diseased tissues (particularly if ER stress is involved in response to the local inflammation).

We believe that increase in Tn levels does reflect simply a change in glycosylation but is a marker of activation of a specific regulatory pathway involved in ECM degradation. It does not mean of course that other glycosylation events, involved in other regulatory processes may not also be activated, but this question is not the focus of our study.

To answer the reviewer's comment, we have performed additional staining with different lectins (DSL, ECL and MALII), which recognize various N- and O-glycans but have not observed any specific increase (Supplementary Fig. 1b).

Fig. 1g—Given that the authors propose that Calnexin is O-glycosylated by GALNT2, VVL would then be recognizing Calnexin (indeed, some overlap of VVL and Calnexin staining can be seen in the UT sample). Therefore, using Calnexin as a marker for the ER to show VVL ER localization becomes problematic. Moreover, this study also posits that O-glycosylation of calnexin causes it to move to the cell surface. Therefore a different marker for the ER should be used.

This is a similar point as above and we have switched ER marker for PDIA3 in Figure 1g and 1h.

Fig. 1h—Given the resolution of these images and the fact that a Golgi marker has not been included, it is impossible to conclude that GALNT2 has moved from the Golgi to the ER. In fact, it looks like there has been a dramatic increase in the expression of GALNT2 and a dramatic expansion of calnexin staining (which because of its putative relocation to the cell surface may not just be staining the ER), potentially indicative of ER stress.

We have added data in Supplementary Fig.1c, d, e indicating that increasing levels of GALNT2 expression does not result in increase of Tn. This point was also reported before in our 2013 publication. Mechanistically, it seems that Tn can only significantly increase if a fraction of GALNTs are relocated to the ER or if the Tn elongation enzymes (C1GALT) is disabled.

We have also added colocalisation data of GALNT2 with PDIA3 and TGN46 in Figure 1h, Figure 2f and g.

Fig. 2d—This experiment is used to support a role for cytokines in increasing matrix degradation in subsequent experiments (in Fig. 3). However, the groups that are being compared do not address whether or not cytokines increase HPL intensity in OASF or RASF cells (e.g. OASF untreated vs OASF cyto, etc....).

We believe there might be a slight misunderstanding on the part of the reviewer. Figure 2d does show unstimulated OASF and RASF (UT conditions). We have added a significance statistical test on this data.

Fig. 2e—This is very difficult to distinguish the color for HPL and Giantin here. Again, both ER and Golgi markers should be included to quantitate the degree of overlap with each compartment.

Thanks for this remark, it was indeed hard to visualize the change on the small format. We have repeated the experiment and used grey scale images to illustrate the changes. We have also added the Figure 2f to show the colocalisation with ER and Golgi markers.

Fig. 4—Is the increase in Calnexin seen via FACS due to increased expression of Calnexin or a change in the cell surface localization? Do the authors have high magnification views of untreated and treated cells stained for calnexin, another ER marker and a cell surface marker to assess changes in calnexin localization?

The change is due to an increase in cell surface expression, since cells were stained non permeabilised. We have not observed any significant change in the total amount of Calnexin.

Rebuttal re manuscript NCOMMS-21-22464A

We thank the reviewers for their contribution to the per-review process and much appreciate that they allocated time to review our manuscript.

Reviewers' comments:

Reviewer #1 (Remarks to the Author):

The authors have addressed my comments.

Thanks.

Reviewer #3 (Remarks to the Author):

The authors state the following as the major conclusions of their paper (last paragraph in the Introduction) which are largely unsupported by direct data. This paper therefore does not meet the standards of publication for Nature Communications:

“In this report, ...(citation of the manuscript)

The authors have failed to demonstrate that the GALNTs are relocated to the ER in patient samples. The authors state that the GALNT2 antibody did not work in patient samples. Therefore, the conclusion that GALNT2 relocates to the ER in arthritic patients cannot be made. As the authors are aware, lectin staining (lectins recognize glycans attached to proteins) cannot be used as a surrogate for the GALNT enzymes as many glycosylated proteins are known to move throughout the cell via anterograde transport, retrograde transport, endocytosis, etc.....after being glycosylated. VVL/HPL staining are detecting (likely) hundreds of different proteins that have had a GalNAc added to them and may be retrotransported to the ER. This has no bearing on where the enzymes that glycosylate them are located.

With this line of reasoning, the reviewer implies that there would be a mechanism by which Tn-bearing proteins are “retrotransported” more actively in arthritic synovial fibroblasts than in resting fibroblasts. To our knowledge there has never been such a mechanism proposed. By contrast, the relocation of GALNTs from Golgi to ER has been amply documented in multiple publications (please see our manuscript for references). Specifically, we have shown that GALNTs are exported from the Golgi in a Src dependent fashion in JCB 2010 (1) and that Src phosphorylates GBF1, a known regulator of retrograde traffic and critical regulator of GALNTs export from the Golgi (2). We have demonstrated previously that GALNTs relocation results in an increase of Tn in the ER. In addition, in this study, we show that the ER-resident protein Calnexin becomes Tn-glycosylated and remains largely located in the ER. This is inconsistent with the alternative mechanism proposed by the reviewer.

Additionally, in instances where the authors looked at GALNTs directly in mouse tissues, costaining with both a Golgi and an ER marker should be done to conclusively demonstrate that the GALNTs have moved from the Golgi to the ER. While they did include an ER marker, it is again one that is purported to form a complex with the

hyperglycosylated calnexin (PDIA3) in the experimental state and cannot be used as a true ER marker.

As we have mentioned in the text, most of the pool of ER resident proteins such as Calnexin and PDIA3 do not actually leave the ER. This is apparent in the staining pattern obtained with both these proteins, which remains largely cytoplasmic and reticulated as is typical of the ER. We have also reported this aspect in our previous publication in *Nature Cell Biology* (3).

In the instance where they looked at GALNT directly in cells with a Golgi and ER marker, it appears that most of the GALNT staining is still in the Golgi (Fig. 2f). So this is clearly not as simple as “re-compartmentation” of the GALNTs to the ER, as most staining remains in the Golgi.

It is indeed important to not take a simplistic view of a population of proteins behaving “en bloc” and all switching from the Golgi to the ER. As we have stated in the text and documented in previous publications, the relocation process is not an “on-off” switch but a gradual process, with variable portion of the Golgi pool of GALNTs being relocated. This can explain in turn the variable levels of Tn seen in the arthritic tissues from patients. Since the Golgi is a smaller and more compact organelle than the ER (estimated to be 10x larger and known to form a meshwork filling most of the cytoplasm), the Golgi pool of GALNTs is always more apparent and easily detectable than the ER pool.

The second conclusion is that glycosylated calnexin is translocated to the surface of the cells and is responsible for the ensuing ECM degradation and disease progression is not adequately supported with direct data. Is the cell surface calnexin specifically hyperglycosylated as their model predicts? With stimulation, what percentage of calnexin is hyperglycosylated and moved to the cell surface? In their rebuttal, the authors state that calnexin can be used as a marker for the ER as most of the calnexin remains in the ER even under stimulated conditions. What does that mean in terms of this model?

Again, as for the GALNTs, it is important to avoid thinking in binary terms: the whole pool of Calnexin does not switch from one location (ER) to the other (surface). The two pools have very different functions for cells and therefore have also different sizes.

At the cell surface, Calnexin forms with PDIA3 an enzymatic complex that concentrates in invadosomes, highly dynamic structures that tend to be labile (4). The effects of this surface localisation can be physiologically deleterious as the ECM structure of tissues is affected. It is thus not highly surprising that cells tightly regulate the proportion of Calnexin that reaches the cell surface and that this proportion is relatively modest. On the other hand, from a therapeutic viewpoint, this is a rather good news as therapeutic antibodies will be in large molecular excess.

Finally, this paper implies that, based on the unsupported conclusions above, a therapeutic option for OA and RA patients may be antibodies to calnexin. Given these unknowns, and the unknown mechanism by which the ECM is actually being degraded, it is difficult to interpret what antibodies to calnexin might be doing in vivo. Are these antibodies specific to the hyperglycosylated form of calnexin?

The antibodies are not specific to glycosylated Calnexin. The specificity to synovial fibroblasts or high GALA cells is brought on by the surface translocation of Calnexin. In terms of mechanism, we have clearly documented (Figure 3a, b) that the anti-Cnx antibodies block ECM degradation, providing a clear and easily understandable therapeutic mechanism.

What effect might they be having on the regular functions of calnexin? Are they influencing secretion or protein folding of other factors, such as proteases?

As the regular function of Calnexin is in the ER, it is shielded from antibodies, which will typically not reach the ER of cells.

Are they influencing immune cells? Given the published roles for calnexin in respiration and T cell migration, intraperitoneally injected calnexin antibodies could be having a plethora of effects on immune system function unrelated to the proposed mechanism here. We should be mindful not to overstate the potential therapeutic relevance without a thorough investigation of the mechanism. Indeed, OA and RA are very complex and distinct diseases, with unique pathologies, immune system involvement, risk factors and progressions.

The reviewer is correct of course that there is work to do before an anti-Calnexin antibody can be utilised as a therapeutic. Ours is a POC study and it would require a lot of detailed work to make sure targeting Calnexin is safe for patients and whether it would be beneficial for OA or RA patients. Our argument is nonetheless very straightforward: arthritis (both OA and RA) is characterised by cartilage degradation, this is driven by synovial fibroblasts, which activate surface Calnexin exposure to induce cartilage ECM cross-linking reduction and promote degradation.

There are currently no data to indicate that any of the GALNTs undergo “re-compartmentation” in either OA or RA patients.

We beg to differ as discussed above. We have provided ample data documenting the relocation of GALNTs in RA synovial fibroblasts.

Reviewer #4 (Remarks to the Author):

The authors have performed substantial work to extend and support their study, however, extrapolation from the human immune staining still overreaches the data presented and should be discussed in a more balanced way.

‘Human tissue microarray (Catalog number 401 3201) of synovial tissues was obtained from provitro AG (Berlin, Germany). Patients’ information were provided in the datasheet available on the company’s website’.

The datasheet on the company’s website shows only limited data from 28 tissue spots – not the complete data from all 52 samples assessed. The clinical data for all of the tissue stained must be included as a supplementary table. Information including age, tissue biopsy location, gender, disease duration, DAS, ESR/CRP, tissue erosion, and treatment would be useful. Does GALA activation correlate with any of these metadata? For example if there is no association with markers of inflammation but with the numbers of erosions this would support a link between increased O-glycosylation and tissue destruction, rather than inflammation.

After verification, indeed, Provitro publishes only 28 tissue spots under the catalog number 401 3201 on their website. However, they have prepared various lots of TMA with different sets of patient samples under the same catalog number i.e. different lot numbers. We have obtained 2 TMA sets with lot numbers 314P210915.22-44 (36 spots) and

105P230819.slide54 (28 spots). We managed to quantify VVL staining of 52 cores out of the supposed 64 samples as some of the annotated cores were missing on the slide or are too small to make out the synovial membrane area.

We have added a table (Table 1) with the clinical data obtained from Provitro. This table includes multiple information regarding the patients and a synovitis score for the biopsy samples. We have not found any correlation with the available parameters (slides attached). We have not added a panel given this lack of obvious correlation.

There is also no clear correlation between Tn levels and the synovitis score. However, this is not very surprising since synovitis score is linked to inflammation and is usually much higher in RA than in OA. Synovitis appears highly linked to immune cells and their influx in the joint. Tn increase in Synovial Fibroblasts appears therefore to be an orthogonal marker for arthritis, possibly more directly related to the ECM degradative activity within the joint.

We added the following text: Line 130

Interestingly, OA samples were more variable than RA. Analysis of patient data did not reveal any correlation between the clinical information available and the Tn levels (Table 1).

“In patient samples, Tn is detectable in all samples of OA, RA and psoriasis arthritis, but the levels are variable. This could be explained if GALA is only fully activated during the active ECM degradative phase of the disease, a phase that would correspond to flares in patients. By contrast, in phases of remission, there would be less ECM degradation and correspondingly lower glycosylation levels”.

Whilst it is true that the heterogeneity in Tn levels could be due to disease in flare or remission there are no human data to support this in this manuscript and it is difficult to make firm conclusions about human disease from the CAIA model. It is also true that the differences observed could be due to any number of other clinical features including but not limited to disease activity, disease duration and/or treatment response. The groups of OA and RA examined here are relatively small and a much larger study with better defined patient cohorts and more detailed clinical metadata are needed to be able to support any conclusion about human disease staging. This lies beyond the current study but the conclusions on page 14 must be amended and a more balanced discussion added. The need for better stratification of patients in much larger independent cohorts should also be discussed.

We agree with the reviewer that the differences in Tn levels could be due to a variety of reasons. We have amended the text in the following way: (Line 405)

“In patient samples, Tn is detectable in all samples of OA, RA and psoriasis arthritis, but the levels are variable. This could have various causes, including individual clinical differences. An interesting hypothesis that would need to be explored further is that GALA might only be fully activated during the active ECM degradative phase of the disease, a phase that could correspond to flares in patients. By contrast, in phases of remission, there is probably less ECM degradation and correspondingly lower glycosylation levels. Overall, it will be important to expand the range of patient samples being analyzed and test whether some clinical parameters such as rate of disease progression might correlate with Tn levels. ”

1. Gill DJ, Chia J, Senewiratne J, Bard F. Regulation of O-glycosylation through Golgi-to-ER relocation of initiation enzymes. *J Cell Biol.* 2010 May 31;189(5):843–858. PMID: PMC2878949
2. Chia J, Wang SC, Wee S, Gill DJ, Tay F, Kannan S, Verma CS, Gunaratne J, Bard FA. Src activates retrograde membrane traffic through phosphorylation of GBF1. *Elife* [Internet]. 2021 Dec 6;10. Available from: <http://dx.doi.org/10.7554/eLife.68678> PMID: PMC8727025
3. Ros M, Nguyen AT, Chia J, Le Tran S, Le Guezennec X, McDowall R, Vakhrushev S, Clausen H, Humphries MJ, Saltel F, Bard FA. ER-resident oxidoreductases are glycosylated and trafficked to the cell surface to promote matrix degradation by tumour cells. *Nat Cell Biol.* Nature Publishing Group; 2020 Nov;22(11):1371–1381. PMID: 33077910
4. Linder S, Wiesner C, Himmel M. Degrading devices: invadosomes in proteolytic cell invasion. *Annu Rev Cell Dev Biol.* 2011 Jul 21;27:185–211. PMID: 21801014

Does Tn levels have any correlation with any clinical profile of the patients?

Reviewer #4

Background of TMA cores

- Cat no. 401 3201
- 2 lots: 314P210915.22-44 (36 cores) and 105P230819.slide54 (28 cores)
- 51/64 samples assessed for Tn levels (VVL intensity/ nuclei intensity):
 - 6 normal
 - 18 RA
 - 22 OA
 - 5 PSA
- Clinical data available:
 - Synovitis score
 - Age
 - Tissue
 - Diagnosis
 - Gender

Poor correlation between Tn levels and Synovitis score

XY data: Correlation of VVL-syn score

Correlation test:

Correlation		A
Tabular results		Score
1	Pearson r	
2	r	0.03810
3	95% confidence interval	-0.2400 to 0.3104
4	R squared	0.001451
6	P value	
7	P (two-tailed)	0.7907
8	P value summary	ns
9	Significant? (alpha = 0.05)	No
10		
11	Number of XY Pairs	51

Linear regression:

Best-fit values	
Slope	0.004770
Y-intercept	0.4536
X-intercept	-95.11
1/slope	209.7
Std. Error	
Slope	0.01787
Y-intercept	0.08549
95% Confidence Intervals	
Slope	-0.03115 to 0.04069
Y-intercept	0.2818 to 0.6254
X-intercept	-infinity to -7.211
Goodness of Fit	
R squared	0.001451
Sy.x	0.2790
Is slope significantly non-zero?	
F	0.07122
DFn, DFd	1, 49
P value	0.7907
Deviation from zero?	Not Significant

No obvious trend between synovitis scores and Tn levels

- No obvious correlation between synovitis score and Tn levels in RA and OA. Too few PSA samples to make any conclusions.
- Noticed higher Tn score at score of 2-3. Perhaps Tn increase at earlier stage and decline at later stage of disease?
- Synovitis scoring for RA patients tend to be higher than OA due to the inflammatory component (Krenn, 2002).
 - 0 to 1 corresponds to no synovitis (inflammatory grade = 0)
 - 2 to 3 to a slight synovitis (inflammatory grade 1)
 - 4 to 6 to a moderate synovitis (inflammatory grade 2)
 - 7 to 9 to a strong synovitis (inflammatory grade 3).
- The score ranges are different between diseases in these cores i.e, RA ranges 3-8 while OA ranges 2-6. More inflammation does not seem to affect Tn levels?

Some correlation observed between Tn levels with age of patients

Tabular results x XY data x | v |

Correlation		A
Tabular results		Age
1	Pearson r	
2	r	0.3833
3	95% confidence interval	0.1204 to 0.5959
4	R squared	0.1469
5		
6	P value	
7	P (two-tailed)	0.0055
8	P value summary	**
9	Significant? (alpha = 0.05)	Yes
10		
11	Number of XY Pairs	51

Correlation due to more samples in the older age group

- There are more OA patient samples (16/22 OA samples) from >60 age range, skewing data.
- No obvious correlation of Tn levels to age in RA and PSA samples

No correlation between Tn levels with tissue location

No correlation between Tn levels with gender

Unpaired t test		
Tabular results		
1	Table Analyzed	WL-Gender_RA-OA-PSA
2		
3	Column B	Female
4	vs.	vs.
5	Column A	Male
6		
7	Unpaired t test	
8	P value	0.3574
9	P value summary	ns
10	Significantly different (P < 0.05)?	No
11	One- or two-tailed P value?	Two-tailed
12	t, df	t=0.9302, df=43
13		
14	How big is the difference?	
15	Mean of column A	0.4482
16	Mean of column B	0.5326
17	Difference between means (B - A) ± SEM	0.08438 ± 0.09070
18	95% confidence interval	-0.09854 to 0.2673
19	R squared (eta squared)	0.01973
20		
21	F test to compare variances	
22	F, DFn, Dfd	1.069, 31, 12
23	P value	0.9477
24	P value summary	ns
25	Significantly different (P < 0.05)?	No
26		
27	Data analyzed	
28	Sample size, column A	13
29	Sample size, column B	32
30		

Reviewer comments in black, reply in blue

Reviewer #3:

It appears as though the authors have replied to my previous critique by citing previously published studies from their group, rather than performing additional analyses or presenting new data. The conclusions of this paper should be based on the data within this paper, not on previously published studies in other systems. Given that there appears to be no new data or experiments, I will reiterate my primary concerns.

The authors have failed to demonstrate that the GALNTs are relocated to the ER in patient samples. The authors state that the GALNT2 antibody did not work in patient samples. Therefore, the conclusion that GALNT2 relocates to the ER in arthritic patients cannot be made.

We apologise for there seems to be some miscommunication with the reviewer. We hold that we have demonstrated the relocation of GALNTs in arthritic conditions. We do provide several pieces of data showing the relocation of GALNT2 in arthritic samples as explained below.

As a reminder, we show high Tn levels in patient samples (Fig.1a) and in mouse tissue (Fig 1e) with an ER pattern (Fig 1g).

These results are very similar to previous published reports. We have documented several times in these reports that Tn staining in the ER is due to GALNT activity. This was shown by siRNA depletion of GALNTs. Expression of an ER-targeted GALNT reproduce ER-localised high Tn.

To illustrate with one example: the change in localisation and increase in levels of Tn is documented in Nguyen et al, 2017, Cancer Cell:

[figure redacted]

In this figure from Nguyen et al., Tn staining revealed by VVL clearly increases and adopts an ER pattern. We also show in this paper the relocation of GALNTs.

In addition, we show directly in this manuscript that GALNT2 in mouse samples is co-localising with an ER marker after induction of arthritis (Fig 1h).

While the staining was a bit challenging initially, we have since added GALNT2 staining in patient samples. Fig S2D show an ER pattern for GALNT2 in arthritic patients samples, with an ER pattern present in lining synovial fibroblasts and a Golgi pattern observable in more deeply seated cells, where the staining co-localises with the Golgi marker TGN46 (Supplementary Fig 2D).

Fig S2D: GALNT2 in patient samples presents an ER pattern at the synovial membrane edge. GALNT2 does not colocalize with the Golgi marker in purple (arrowhead). By contrast, cells in a deeper setting (not the edge) show GALNT2 colocalising with the Golgi marker TGN46 (arrow)

In addition, in synovial fibroblasts extracted from surgical samples from human arthritic patients, we found that synovial fibroblast cells have an increased GALNT activity with a typical ER pattern (Fig 2.e).

In Figure 2e of this manuscript: the pattern for Tn in stimulated cells is ER-like.

We also show that, in a synovial fibroblast line, that GALNT2 shows a typical ER pattern, localising with the ER marker PDIA4 after stimulation by cytokines and cartilage ECM (Fig 2f).

Similar staining pattern and relocation of GALNT to the ER was also amply documented in several of our publications, including Chia et al (PLOS One, 2019). For instance in Figure 2A, one can observe dispersion of GALNT1 staining after EGF stimulation and after ERK8 depletion:

“Additionally, in instances where the authors looked at GALNTs directly in mouse tissues with an ER marker, they used Calnexin as the ER marker, which is supposed to be the protein that is mis-located to the cell surface in this model. The authors respond that the

GALNT relocation to the ER is not “on-off” but is variable, with most of the GALNTs remaining in the Golgi. Likewise, they respond that the majority of Calnexin remains in the ER, so it can be used as an ER marker. This raises concerns as to how this model can be rigorously tested if it is unclear to what extent mislocation occurs and how this does or does not correlate with phenotypes observed. How much GALNT in the ER is needed to cause the effects? How much Calnexin at the cell surface is abnormal? What percentage of relocation of the GALNTs and Calnexin is necessary to see the phenotypes? Is all calnexin within the ER glycosylated normally and all on the cell surface abnormally glycosylated?”

Here, we find the argument rather convoluted. We do not claim that the relocation of GALNTs to the ER and/or ER proteins to the cell surface must be complete in to have an effect. We also do not think it needs to be in order to be “rigorously tested”.

We have previously reported that the relocation of GALNTs does not need to be complete to have a massive effect on Tn levels. We discussed it in “Regulation of O-glycosylation through Golgi-to-ER relocation of initiation enzymes.” Gill et al. JCB 2010, as well as in “Initiation of GalNAc-type O-glycosylation in the endoplasmic reticulum promotes cancer cell invasiveness.” Gill et al. PNAS 2013 and “The GalNAc-T Activation (GALA) Pathway: Drivers and markers” Chia et al. PLOS One 2019.

In the paper “ERK8 is a negative regulator of O-GalNAc glycosylation and cell migration” by Chia et al. eLife, 2014, we have shown that ERK8 depletion leads to massive Tn increase (Fig 1D). In these conditions, GALNT1 relocation is partial (Fig 5E).

Fig 1D: ERK8 depletion leads to massive Tn increase and Tn ER pattern

Figure 5 E: GALNT1 relocation.
 In this panel, one can clearly see that GALNT1 relocation from Golgi to ER is not complete

Similarly, regarding the relocation of the ER resident protein Cnx from ER to the cell surface, we have not claimed that the relocation was complete. In “ER-resident oxidoreductases are glycosylated and trafficked to the cell surface to promote matrix degradation by tumour cells” by Ros et al, Nature Cell Biology 2020.

We show partial relocation of Calnexin from the ER to the surface. The photo chosen by the journal as a cover picture display this partial relocation:

In green is the staining for Calnexin, which is largely intracellular in the ER but colocalizes partially with actin in actin rings in red.

[image redacted]

Given the lack of data for GALNT relocation in patient samples and the lack of correlation between Tn levels and synovitis score, it would be premature to suggest antibodies to calnexin as a potential therapy.

In this last paragraph, an argument is presented about the correlation between synovitis score and Tn levels being essential for publication. We never claimed there was such a correlation. Indeed, synovitis score is highly dependent on immune inflammation, i.e. the immune cells involved in RA. Our study is about the stromal component, the synovial fibroblasts. There are interactions between the immune cells and the fibroblasts but that does not mean that there is a strict correlation at the time of biopsy between synovitis score and fibroblasts GALA activation.

In summary, we believe we have amply demonstrated in this manuscript that GALNTs relocate to the ER in synovial fibroblasts that drive cartilage degradation. In addition, the data presented and the model proposed is in line with concepts described in previous publications.

References:

Nguyen AT, Chia J, Ros M, Hui KM, Saltel F, Bard F. Organelle Specific O-Glycosylation Drives MMP14 Activation, Tumor Growth, and Metastasis. *Cancer Cell*. 2017 Nov 13;32(5):639–653.e6. PMID: 29136507

Ros M, Nguyen AT, Chia J, Le Tran S, Le Guezennec X, McDowall R, Vakhrushev S, Clausen H, Humphries MJ, Saltel F, Bard FA. ER-resident oxidoreductases are glycosylated and trafficked to the cell surface to promote matrix degradation by tumour cells. *Nat Cell Biol*. Nature Publishing Group; 2020 Nov;22(11):1371–1381. PMID: 33077910

Gill DJ, Chia J, Senewiratne J, Bard F. Regulation of O-glycosylation through Golgi-to-ER relocation of initiation enzymes. *J Cell Biol*. 2010 May 31;189(5):843–858. PMCID: PMC2878949

Gill DJ, Tham KM, Chia J, Wang SC, Steentoft C, Clausen H, Bard-Chapeau EA, Bard FA. Initiation of GalNAc-type O-glycosylation in the endoplasmic reticulum promotes cancer cell invasiveness. *Proc Natl Acad Sci U S A*. 2013 Aug 20;110(34):E3152–61. PMCID: PMC3752262

Chia J, Tay F, Bard F. The GalNAc-T Activation (GALA) Pathway: Drivers and markers. *PLoS One*. 2019 Mar 19;14(3):e0214118. PMID: 30889231

Chia J, Tham KM, Gill DJ, Bard-Chapeau EA, Bard FA. ERK8 is a negative regulator of O-GalNAc glycosylation and cell migration. *Elife*. 2014 Mar 11;3:e01828. PMCID: PMC3945522

REVIEWER COMMENTS

Reviewer #5 (Remarks to the Author):

In this study, the authors used human samples, mouse models, and different cell lines to test the role of O-glycosylation in ECM remodeling in arthritic diseases. They provided evidence that Golgi-localized GALNTs are relocated to the ER in activated arthritic synovial fibroblasts (SFs), leading to increased glycosylation and cell surface localization of the ER chaperone Calnexin. Calnexin participates in matrix degradation by reducing ECM disulfide bonds; anti-Calnexin antibodies block ECM degradation and protect animals from arthritic diseases. Altogether, the authors conclude that ER O-glycosylation in synovial fibroblasts drives cartilage degradation. Overall, the topic is interesting, it reveals a novel role of O-glycosylation in ECM remodeling in arthritic diseases.

We thank the reviewer for his/her comments and appreciate her thorough evaluation of our manuscript.

Major concerns:

1. There is a lack of solid evidence to support the ER localization of GALNTs in activated arthritic synovial fibroblasts. Figure 1ab (and following figures), using the VVL signal to represent the localization of GALNT is indirect, as many GALNT substrates are supposed to be transported from the ER to other cellular locations.

Staining directly for GALNTs in tissues has been technically challenging in the past, in part because the antibody we used tended to aggregate. To address this problem, we performed serial sectioning of an RA patient knee sample, then did multiple stainings: VVL/Nuclei, GALNT2/TGN46/nuclei and GALNT2/CRT/nuclei. The images obtained are displayed in Fig 1c. The panel shows high-resolution images, with a layer of cells with high Tn levels (VVL staining) at the periphery and a zone of low Tn levels inside the tissue. At the periphery, GALNT2 displays a diffuse pattern and colocalizes with the ER marker, while in the center GALNT2 colocalizes with the Golgi marker TGN46.

The added text reads as follows:

We investigated the intracellular localization of GALNT2, as the translocation of GALNTs to the ER has been linked to increased Tn levels (Gill et al., 2010; Gill et al., 2013; Chia et al., 2014; Chia et al., 2019). To this end, we performed serial sectioning on a knee synovium sample from an RA patient, labeled the sections with specific markers, and conducted high-resolution confocal microscopy. Our analysis revealed that cells with high Tn levels were concentrated at

the edge of the synovium, forming a layer only 2-3 cells thick (Figure 1c). In these cells, GALNT2 colocalized with the ER marker Calreticulin, exhibiting a distinct ER localization pattern (Figure 1c). In contrast, cells deeper in the synovium showed lower Tn levels and GALNT2 colocalized predominantly with the Golgi marker TGN46 (Figure 1c). These findings indicate that, in arthritic synovium, cells at the periphery relocate GALNT2 from the Golgi to the ER, leading to a pronounced increase in intracellular Tn levels.

In addition, the VVL signal in RA and OA samples is in average 3-fold of control in Figure 1b, not seven-fold as the authors claimed.

There is a misunderstanding, our text read as “While healthy subjects samples showed little variation, most samples of RA and OA samples displayed enhanced Tn levels, with in some areas with up to seven-fold increase of VVL signal (Figure 1b).”

For clarity, we replaced it with: “While healthy subjects samples showed little variation, enhanced Tn levels were observed in most samples of RA (average 2 fold increase) and OA samples (average 3 fold increase, with some samples displaying seven-fold increase) (Figure 1b).”

Supplementary Fig. 1c, the GFP signal in the Golgi-G1 cells is mostly in the ER, not Golgi. It is unclear why the HPL signal is not increased in Golgi-G1 cells as in the ER-G1 cells regardless of the localization of the enzyme.

In Supplementary Fig. 1c, we use bicistronic constructs and GFP is separated from GALNT1 sequences by a T2A sequence, as described in Korwarz E, et al. Optimized Sleeping Beauty transposons rapidly generate stable transgenic cell lines. *Biotechnol J.* 2015 Apr;10(4):647-53. Therefore, GFP is not localized at the Golgi. We have added a western blot where we measure the levels of exogenous GALNT1 using the V5 tag. This blot demonstrate that both enzymes are expressed at similar levels and only ER-G1 is able to raise Tn levels.

The fact that Tn levels are not increased by over-expression of wild type GALNT1 (Golgi-G1) but only by ER-targeted GALNT1 (ER-G1) is the point we are making: GALNTs localization and not their level of expression influence Tn levels.

Corresponding text:

We showed previously that GALNT1 and 2 are relocalised conjointly and similarly increase Tn levels (Gill et al. 2010). To further demonstrate that subcellular location and not expression levels of these enzymes regulate Tn levels in synovial fibroblasts, we compared the effect of over-expression of ER-localised versus Golgi-localised GALNT1. Using available plasmids, we transformed human synovial fibroblastic SW982 cells with GFP-expressing bi-cistronic constructs with either wild-type GALNT1 (Golgi-G1) or a GALNT1 fused to an ER-retention sequence (ER-G1) as previously described (Supplementary Fig. 1d) (Gill et al. 2013; Kowarz et al. 2015). All three cell lines had comparable GFP level expression (Supplementary Fig. 1e). By contrast, Tn levels measured with HPL staining showed a significant, 3-fold increase in ER-G1 expressing cells while GFP and Golgi-G1 cells were unaffected (Supplementary Fig. 1f) (Gill et al. 2013).

Tn is a transient glycan in the Golgi, being rapidly modified by glycan extension enzymes. The levels at the Golgi are already at saturation when Golgi-G1 expression is added. By contrast, the ER-localized enzyme accesses a large pool of unmodified substrates and Tn is not modified in the ER, leading to a large increase in Tn and not more extended glycans (like T).

We added this text in the Discussion:

Indeed we find that only relocation and not overexpression of GALNTs significantly affect Tn levels in synovial fibroblasts. A likely explanation is that relocated GALNTs modify new substrates such as ER-resident proteins and that these Tn glycans are not capped by enzymes such as C1GALT²⁷.

Supplementary Fig. 2d, while the authors showed that the GALNT2 signal in synovial fibroblasts at the forefront exhibited diffused, ER-like staining, but they did not provide the result in control tissues/cells for comparison.

Please refer to the new figure 1c. We show the colocalisation of GALNT2 with ER marker in cells at the synovium margin. Deeper in the synovium, GALNT2 displays the expected Golgi-like pattern, constituting a beautiful internal control of localisation. In addition, we show that in normal joints, Tn is not elevated.

Supplementary Fig. 3a, it seems that the HPL signal is mostly in the Golgi in almost all cells. GALNT localization is not shown in this figure.

Supplementary Fig 3a,b is shown to demonstrate that ER-2LEC reduces the increase in Tn levels induced by cytokines in Synovial fibroblasts. Thus, we have measured Tn staining as detected by HPL staining. In the lower left-hand image (CYTO+ECM stimulated SW982), the Tn pattern is clearly no longer Golgi, but Golgi + ER, which is typical of GALA activation.

Figure 2f, depending on the experimental procedure (such as the blocking method and the concentration of the antibody used) and the background setting on the microscopy, this result may vary. This important result should be validated by other methods.

In this experiment, we demonstrate that GALNTs are relocated to the ER following stimulation by cytokines and cartilage ECM. The relocation of GALNT2 to the ER recapitulates the observation shown in Fig 1c.

We have treated the cells from both conditions in the same experiment and staining methods. We have acquired the cells under the exact same microscopy conditions. We agree that this is an important result, and we have already published very similar results, with extensive validation in Gill et al, JCB (2010) and Chia et al, PLOS (2019) (see below some examples).

We have shown in Figure 4a that in these stimulatory conditions (CYTO +ECM), the ER-resident protein Calnexin is hyper-glycosylated, further confirming that GALNTs are relocated to the ER.

[figure redacted]

Figure 2 in JCB, 2010

[figure redacted]

Figure 2, Chia et al, PLOS One 2019

There is also no mechanism for how GALNTs are relocalized to the ER under disease conditions.

The mechanism of GALNTs relocation is not the focus of this manuscript. The reviewer may refer to Gill et al, JCB 2010 for the role of Src and Arf1 and COPI. A follow-up study linked Src with the ARf regulator GBF1 can be found at: Chia et al, 2021^{1,2}. Regarding the relocation induced by arthritis, we show in this manuscript that it is induced by cytokines (IL-1beta and TNFalpha) and by exposure to cartilage ECM.

2. No evidence is provided to support that GALNT localization to the ER is the cause of elevated VVL level or Calnexin trafficking to the plasma membrane in disease models.

We had provided this data in figure S1d, e and f, now completed with S1g.

The expression level of GALNTs appears to vary in different results; evidence is needed to confirm that it is GALNT ER localization, rather than increased expression, is the cause of the observed effects. For example, in Figure 1h, while it is true that the signal in the control appear as puncta-like structures, the signal in the disease model is much stronger, raising a concern of whether the appearing broader localization is due to increased expression. Higher magnification images are needed with similar signal intensities to confirm the localization of the proteins. Adding a Golgi marker may also be helpful in this study. In addition, can the authors determine the expression level of GALNT2 by western blotting in this and other experiments?

We have added the high magnification images illustrating the localisation of GALNT2 in Fig 1c (see above), which addresses the point of change of GALNT localisation in disease tissues.

We have also provided direct experimental evidence in Figure S1d,e,f and g demonstrating that GALNT1 increased expression does not significantly change Tn levels.

To note, we have made this point in previous publication and demonstrated that high Tn levels are not induced by over-expression of GALNTs but through their relocation to the ER³.

How could Calnexin glycosylation in the ER lumen affects its ER retention signal at the cytoplasmic tail? Calnexin cleavage has been reported in the literature, have the authors consider this possibility?

We agree that the effect of glycosylation must be indirect; the precise mechanism is at present unknown and we are working on it.

Many experiments are incomplete, the authors used different systems to show different results, such as VVL signal level in tissues (Fig. 2ab) and GALNT localization in cells (Fig. 2fg), making it difficult to correlate the different results in the same experimental setting.

We hope the new data will assuage this concern. As described above, we have been using the lectins VVL and HPL in several published works. We show direct relocalisation of GLANT2 in Fig 1c.

3. There is no mechanism/evidence of how cell surface location of Calnexin affects ECM degradation. The authors reasoned this as the secretion of PDIA3 but did not provide experimental evidence.

The mechanism of Cnx effect on ECM degradation is not the focus of this paper. It was the focus of a previous paper by M. Ros et al. NCB (2020) where the mechanistic aspect was studied in detail⁴. The paper reports that the complex Cnx/PDIA3 is involved in reducing disulfide bonds in the ECM.

In this manuscript, we provide evidence in Supp Figure 5a-c that cells reduce disulfide bonds in the ECM and that anti-Cnx antibodies block that process.

Figure 3, there is no result shown to correlate the expression level/localization of exogenous and endogenous GALNT1 with ECM degradation.

As mentioned in the introduction, we have shown extensively the link between ER-localisation of GALNTs and increase ECM degradation in two different publications: Nguyen et al. 2017 and Ros et al. 2020^{4,5}.

Figure 4, GALNT1 blot should be shown. Actin blot shows that the loading is uneven, making the result inconclusive.

In figure 4b, the quantification takes into account the slight variation in actin input and corresponds to three different experiments/blots.

Supplementary Fig. 5b, since these are permeabilized cells, why there is no signal in control cells? This raises a concern that increased GALNT expression is more important than its ER-localization.

There might be some misunderstanding there as Sup Fig 5b is about an SCFv binding to Calnexin. There is no question of GALNT expression or localisation in this experiment.

Here is a speculation of how adding anti-calnexin antibodies to cells or animals could reduce ECM degradation: calnexin is a lectin that binds glycoproteins including MMPs, and antibody binding would trigger endocytosis and degradation of cell surface calnexin together with MMPs, leading to the reduction of ECM degradation.

We have demonstrated that knock-down and knock-out of Calnexin reduce or block ECM degradation ⁴. The effect is thus not exclusive to antibodies and the mechanism proposed by the reviewer is inconsistent with the data.

4. The writing needs to be improved. Examples include “autoreactive B and T cells produce autoantibodies” (T cells do not produce antibodies), “the relocation of GALNTs to the ER is a reversible event under the control of signaling pathways” (this is not shown in the paper), and “2x LDS sample buffer” (might be SDS sample buffer?). In addition, abbreviations should be spelled out when appear first time.

We thank the reviewer for pointing out these mistakes, unfortunately inherent to large and complex manuscripts. We have modified the text to remove T- cells and change LDS to SDS. We have added the reference for the reversible relocation: ⁶

1. Gill DJ, Chia J, Senewiratne J, Bard F. Regulation of O-glycosylation through Golgi-to-ER relocation of initiation enzymes. *J Cell Biol.* 2010 May 31;189(5):843–858. PMID: PMC2878949
2. Chia J, Wang SC, Wee S, Gill DJ, Tay F, Kannan S, Verma CS, Gunaratne J, Bard FA. Src activates retrograde membrane traffic through phosphorylation of GBF1. *Elife* [Internet]. 2021 Dec 6;10. Available from: <http://dx.doi.org/10.7554/eLife.68678> PMID: PMC8727025
3. Gill DJ, Tham KM, Chia J, Wang SC, Steentoft C, Clausen H, Bard-Chapeau EA, Bard FA. Initiation of GalNAc-type O-glycosylation in the endoplasmic reticulum promotes cancer cell invasiveness. *Proc Natl Acad Sci U S A.* 2013 Aug 20;110(34):E3152–61. PMID: PMC3752262
4. Ros M, Nguyen AT, Chia J, Le Tran S, Le Guezennec X, McDowall R, Vakhrushev S, Clausen H, Humphries MJ, Saltel F, Bard FA. ER-resident oxidoreductases are glycosylated and trafficked to the cell surface to promote matrix degradation by tumour cells. *Nat Cell Biol.* Nature Publishing Group; 2020 Nov;22(11):1371–1381. PMID: 33077910
5. Nguyen AT, Chia J, Ros M, Hui KM, Saltel F, Bard F. Organelle Specific O-Glycosylation Drives MMP14 Activation, Tumor Growth, and Metastasis. *Cancer Cell.* 2017 Nov 13;32(5):639–653.e6. PMID: 29136507
6. Chia J, Tham KM, Gill DJ, Bard-Chapeau EA, Bard FA. ERK8 is a negative regulator of O-GalNAc glycosylation and cell migration. *Elife.* 2014 Mar 11;3:e01828. PMID: PMC3945522